# Morphological bases of phytoplankton energy management and physiological responses unveiled by 3D subcellular imaging

Clarisse Uwizeye [1], Johan Decelle[1✉], Pierre-Henri Jouneau [2], Serena Flori[1,3], Benoit Gallet [4], Jean-Baptiste Keck [5], Davide Dal Bo[1], Christine Moriscot[4,6], Claire Seydoux[1], Fabien Chevalier[1], Nicole L. Schieber [7], Rachel Templin[7], Guillaume Allorent[1], Florence Courtois[1], Gilles Curien[1], Yannick Schwab[7,8], Guy Schoehn [4], Samuel C. Zeeman[9], Denis Falconet [1✉] & Giovanni Finazzi [1✉]

Eukaryotic phytoplankton have a small global biomass but play major roles in primary production and climate. Despite improved understanding of phytoplankton diversity and evolution, we largely ignore the cellular bases of their environmental plasticity. By comparative 3D morphometric analysis across seven distant phytoplankton taxa, we observe constant volume occupancy by the main organelles and preserved volumetric ratios between plastids and mitochondria. We hypothesise that phytoplankton subcellular topology is modulated by energy-management constraints. Consistent with this, shifting the diatom *Phaeodactylum* from low to high light enhances photosynthesis and respiration, increases cell-volume occupancy by mitochondria and the plastid $CO_2$-fixing pyrenoid, and boosts plastid-mitochondria contacts. Changes in organelle architectures and interactions also accompany *Nannochloropsis* acclimation to different trophic lifestyles, along with respiratory and photosynthetic responses. By revealing evolutionarily-conserved topologies of energy-managing organelles, and their role in phytoplankton acclimation, this work deciphers phytoplankton responses at subcellular scales.

[1] Univ. Grenoble Alpes, CNRS, CEA, INRAe, IRIG-LPCV, Grenoble, France. [2] Univ. Grenoble Alpes, CEA, IRIG-MEM, Grenoble, France. [3] The Marine Biological Association, The Laboratory, Citadel Hill Plymouth, Devon, UK. [4] Univ. Grenoble Alpes, CNRS, CEA, IRIG-IBS, Grenoble, France. [5] Univ. Grenoble Alpes, Laboratoire Jean Kuntzmann, Grenoble, France. [6] Univ. Grenoble Alpes, CNRS, CEA, EMBL, Integrated Structural Biology Grenoble (ISBG), Grenoble, France. [7] Cell Biology and Biophysics Unit, European Molecular Biology Laboratory, Heidelberg, Germany. [8] Electron Microscopy Core Facility, European Molecular Biology Laboratory, Heidelberg, Germany. [9] Institute of Molecular Plant Biology, Department of Biology, ETH Zurich, Zurich, Switzerland. ✉email: johan.decelle@univ-grenoble-alpes.fr; denis.falconet@cea.fr; giovanni.finazzi@cea.fr

Phytoplankton play a critical role in supporting life on Earth. By converting $CO_2$, sunlight and nutrients into biomass and oxygen, unicellular phototrophs are responsible for about 50% of primary productivity[1]. They also contribute to food webs and to the biological $CO_2$ pump in the oceans. Phytoplankton members are ubiquitous in marine and freshwater ecosystems and include prokaryotes and eukaryotes, the latter having acquired photosynthesis capacity up to 1.5 billion years ago through endosymbiotic events[2]. Eukaryotic phytoplankton encompasses a great diversity of lineages (e.g. diatoms, dinoflagellates, haptophytes, chlorophytes, rhodophytes) with different morphologies and sizes (from 0.8 to a few tens of microns)[3]. Although our knowledge on phytoplankton biodiversity and ecological relevance in aquatic ecosystems has greatly improved in the recent years (e.g. ref. [4]), the cellular bases of ecological responses of these unicellular organisms remain undetermined. Moreover, we do not know how flexible the phytoplankton cellular and organellar architecture is when facing environmental changes. This is a critical aspect, as recent works have proposed that phytoplankton physiological responses may rely on specific subcellular features[5,6].

So far, phytoplankton morphological features have been mainly visualized by light microscopy and two-dimensional (2D) electron microscopy studies[7–11], often associated with the assessment of photosynthetic activity[10,12]. High-throughput confocal fluorescence three-dimensional (3D) imaging has been developed to scan, classify and quantify phytoplankton cells collected in different oceanic regions[13]. However, optical microscopy studies have insufficient resolution to reveal cellular ultrastructure, and 2D electron microscopy by definition cannot provide a comprehensive volumetric description of phytoplankton cells and their organelles.

Thanks to the recent development of 3D electron microscopy methods[14–16], 3D reconstructions have been obtained to analyse plant cell division[17], chloroplast biogenesis[18], with emphasis on thylakoids organization[19–23] and algal cell structures[24–28]. Serial block-face electron microscopy (SBEM) has been used to analyse plant subcellular architectures[29–31]. Ion-beam milling was used to prepare thin lamella for imaging by cryo-EM[32], revealing the native architecture of the *Chlamydomonas reinhardtii* chloroplast[6,33,34]. Focused ion beam scanning electron microscopy (FIB-SEM) has been used to reveal the 3D structure of photosynthetic cells with enough resolution (4–10 nm) to investigate their subcellular architecture. This technique has been applied to chemically fixed samples in rice[35,36], *Chlamydomonas*[37,38], in the diatom *Phaeodactylum tricornutum*[39,40], and to cryo-fixed and freeze substituted *Phaeocystis cordata* cells[41]. Cryo-FIB-SEM of high-pressure frozen marine algae such as coccolithophores[42] and dinoflagellates[43,44] has also been used to study biomineralization pathways. However, we still miss comparative studies to reveal evolutionarily conserved topologies in eukaryotic phytoplankton and to highlight possible links between acclimation responses and changes in subcellular architectures.

Here, we applied a FIB-SEM-based workflow to seven monoclonal cultures of different eukaryotic microalgae representing major oceanic phytoplankton lineages and/or model-laboratory microalgae. We generate 3D reconstructions, suitable for quantitative morphometric analysis (surfaces and volumes) of organelles and subcellular structures. Comparative analysis of the different lineages reveals preserved structural characteristics between the different species: conserved cell-volume occupancy by the different organelles and constant volumetric ratios in energy-producing organelles (plastids, mitochondria). These relationships between subcellular compartments related to energy management may represent evolutionarily conserved features responsible for specific physiological responses in phytoplankton. Consistent with this idea, physiological responses of microalgae acclimated to either different light regimes or trophic lifestyles are accompanied by commensurate modifications in the structural features of plastids and mitochondria, as well as in their interactions.

## Results and discussion

**Cellular architectures of phytoplankton**. We reconstructed the 3D cellular architecture of different eukaryotic phytoplankton representatives of ubiquitous taxa and laboratory model organisms: Mammiellophyceae (*Micromonas* RCC 827), Prymnesiophyceae (*Emiliania* RCC 909), Pelagophyceae (*Pelagomonas* RCC 100), Dinophyceae (*Symbiodinium* RCC 4014 clade A), Cyanidiophyceae (*Galdieria* SAG 21.92), Bacillariophyceae (*Phaeodactylum* Pt1 8.6), and Eustigmatophyceae (*Nannochloropsis* CCMP 526) (Supplementary Table 1). Prior to FIB-SEM imaging, culture aliquots were tested for photosynthetic capacity (Supplementary Table 1) to verify their physiological status. Cells were cryo-fixed using high-pressure freezing (to maximize preservation of native structures) followed by slow freeze substitution and resin embedding. FIB-SEM datasets were processed to 3D models using open-access software (see Supplementary Fig. 1 and methods for details). This imaging approach allows a wide range of cell volumes to be quantified, from ca. 2 μm³ in the mamiellophyceae *Micromonas*, to more than 200 μm³ in the dinoflagellate *Symbiodinium*.

We observed both external features of microalgae (e.g. the raphe in *Phaeodactylum*, the flagellum in *Micromonas*, the coccosphere in *Emiliania*, Fig. 1), and the main organelles (Fig. 2: nucleus—blue, plastid—green and mitochondria—red). Other cellular features were observed (grey): storage bodies in *Emiliania*[42,45], carbon-rich structures in *Pelagomonas*[46], large oil bodies in *Nannochloropsis*[47], starch sheaths surrounding the pyrenoids in *Micromonas*[48] and *Symbiodinium*, and vacuoles of different sizes in *Phaeodactylum*[49], *Galdieria* and *Micromonas*[48].

Different shapes were observed for the main organelles. Plastids were cup-shaped in *Galdieria*, *Pelagomonas*, *Emiliania*, lobed in *Symbiodinium*, globular in *Micromonas* and *Nannochloropsis* and elongated in *Phaeodactylum* (Fig. 2b and Fig. 3a). When distinguishable, photosynthetic membranes (thylakoids) were organized in layers of a few stacks, but lacked the clear subdivision into stacked grana and unstacked stromal lamellae observed in vascular plants[50]. The nuclei were spherical/oval in shape and were closely associated with the plastids via the fourth envelope membranes in secondary plastids (i.e. *Phaeodactylum*[40]). Mitochondria were characterized by more variable shapes not only between species but also within cells of the same species (e.g. Supplementary Fig. 2 in the case of *Emiliania*). This diversity probably reflects the dynamic nature of these organelles, which frequently change their shape, undergo dislocations, fusion and fission in eukaryotes[51].

Quantitative analysis indicates that plastids always occupied the largest fraction (15–40%) of the cell (Fig. 2c, Supplementary Data 1) in line with recent estimates in vascular plants[29,35], followed by the nucleus (5–15%) and the mitochondria (2.5–5%). Altogether, these three organelle types (nuclei, plastids and mitochondria) filled a relatively constant fraction (40–55%) of the total cell volume, despite significant differences in the cell volumes of the different phytoplankton taxa (*Symbiodinium* e.g. is around 100 times larger than *Micromonas*). Networks of internal vesicles, the Golgi apparatus, ER, vacuoles and storage compartments (e.g. lipid droplets, starch granules, nutrient storage, etc) and the cytosol occupied the other half with a larger variability in terms of their relative volume occupancy. We interpret this conservation of the organelle volumes and the variability of the other compartments as the signature of evolutionary constraints that preserve essential cellular functions (gene expression, energy production and consumption,

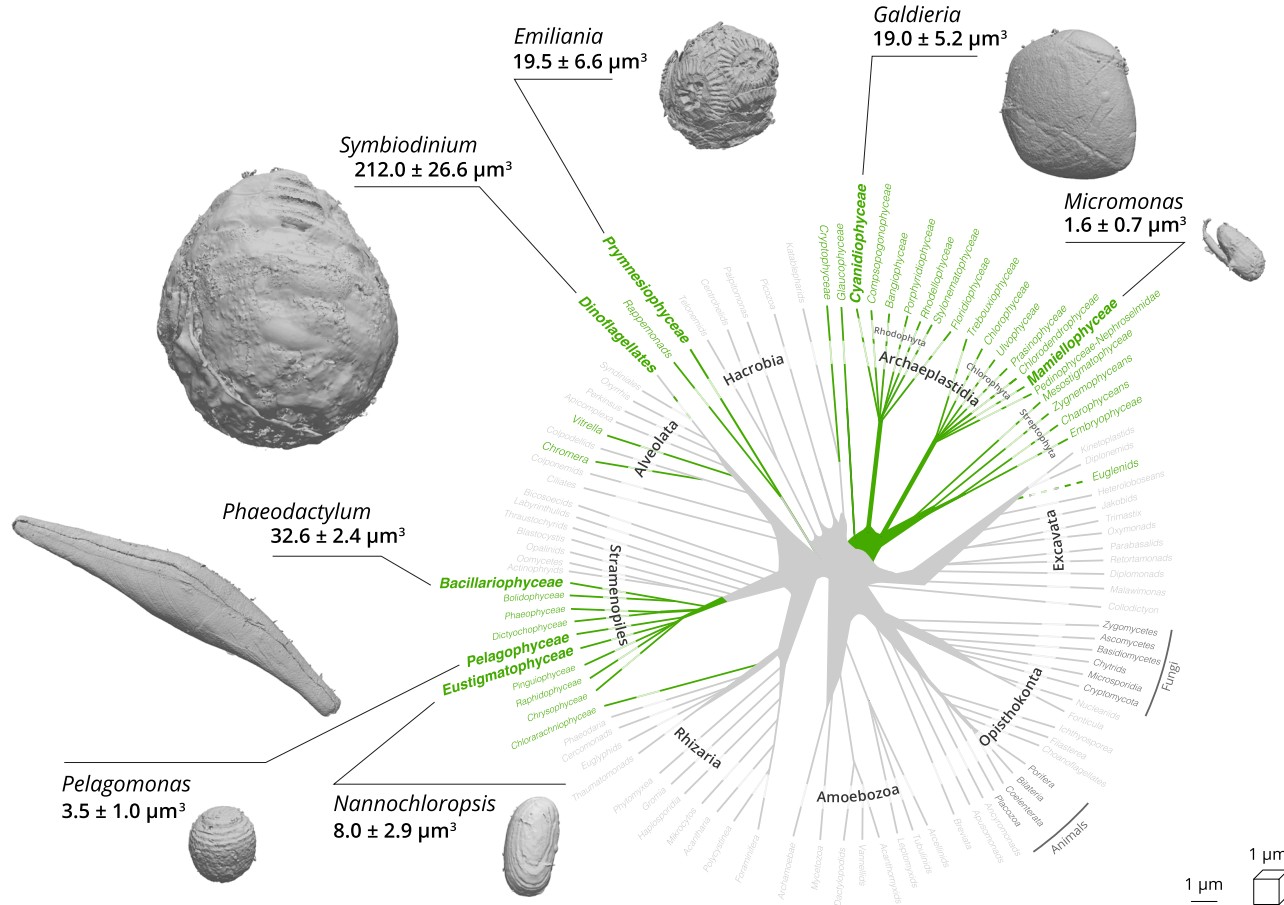

**Fig. 1 Cellular volume and external features of selected phytoplankton cells revealed by FIB-SEM imaging.** Green branches of the phylogenetic tree of eukaryotes represent photosynthetic lineages (adapted from ref. [91]). A 3D scan view of cell morphology of selected phytoplankton members (Mammiellophyceae (*Micromonas* RCC 827), Prymnesiophyceae (*Emiliania* RCC 909), Bacillariophyceae (*Phaeodactylum* Pt1 8.6), Pelagophyceae (*Pelagomonas* RCC 100), Dinophyceae (*Symbiodinium* RCC 4014 clade A), Cyanidiophyceae (*Galdieria* SAG21.92) and Eustigmatophyceae (*Nannochloropsis* CCMP526) is shown with a linear scale bar of 1 μm and a voxel scale of 1 μm³. Specific cellular features (cell walls, the flagellum in *Micromonas*, the raphe in *Phaeodactylum*, the coccosphere in *Emiliania*) are visible. For every species, three cells were reconstructed and morphometrically analysed. Data represent cell volumes ± s.d. for every species.

compartmentation of metabolic pathways), while leaving metabolic flexibility to allow the storage of assimilated nutrients, particularly carbon and subcellular trafficking. The only exception was *Nannochloropsis*, where the large accumulation of oil bodies possibly reduced the cell volume available to the main organelles (22.4 ± 4.5%, Fig. 2, see also below).

Thanks to the possibility to perform quantitative surface and volumetric estimates, we sought relationships between the three above mentioned organelles (Fig. 3) in the different taxa, to identify possible evolutionarily-preserved morphological characteristics. This analysis was initially biased by the presence of *Symbiodinium* (Supplementary Fig. 3). These dinoflagellate cells, being much larger than the others, led to the clustering of data into two groups (*Symbiodinium* cells on one side, all the other cells on the other side), preventing the observation of correlation between the other cells.

Excluding *Symbiodinium* from the analysis removed this bias and unveiled the existence of a tight correlation between plastids and mitochondria in terms of volume (the coefficient of determination $R^2$, being 0.95, Fig. 3b) and surface area ratios ($R^2 = 0.85$, Fig. 3c).

No significant correlation was found between the volume/surface ratio of the nucleus and the mitochondria or plastid ($R^2 ≤ 0.5$). Plastid-mitochondria relationships are of primary importance in

diatoms[5,52], where interactions between the two organelles are relevant for carbon assimilation. Based on the findings above, it is possible that this organelle-organelle relationship also exists in other microalgal species.

Plastid-mitochondria interactions may rely on physical interactions between the two organelles[39,53]. We tested this possibility by quantifying possible contact points between plastids and mitochondria in the different species analysed above (Fig. 4). Recent work based on cryo-electron tomography of cyanobacterial cells has revealed specific contact sites between thylakoids and the plasma membrane with a ~3 nm intermembrane space[54]. Using the same technique, ER-plasma membrane, ER-mitochondria, and nucleus-vacuole contact sites were measured in eukaryotic cells with intermembrane distances of ~20 nm, ~10 nm, and ~15 nm, respectively[55,56]. Based on these results, we chose a distance value of ≤30 nm to calculate surface areas of contact between plastids and mitochondria. We could identify contacts in *Phaeodactylum* (7.1 ± 1.1% of the plastid surface being involved in contacts with mitochondria, Fig. 4a), in agreement with the previous suggestions[5]. Conversely, contacts turned out to be almost negligible in all the other organisms, ranging from 0.1 ± 0.1 in *Pelagomonas* to 1.8 ± 0.8% of the plastid surface in *Emiliania*.

Other distance criteria have been proposed to operationally track contact points between organelles in light microscopy[57].

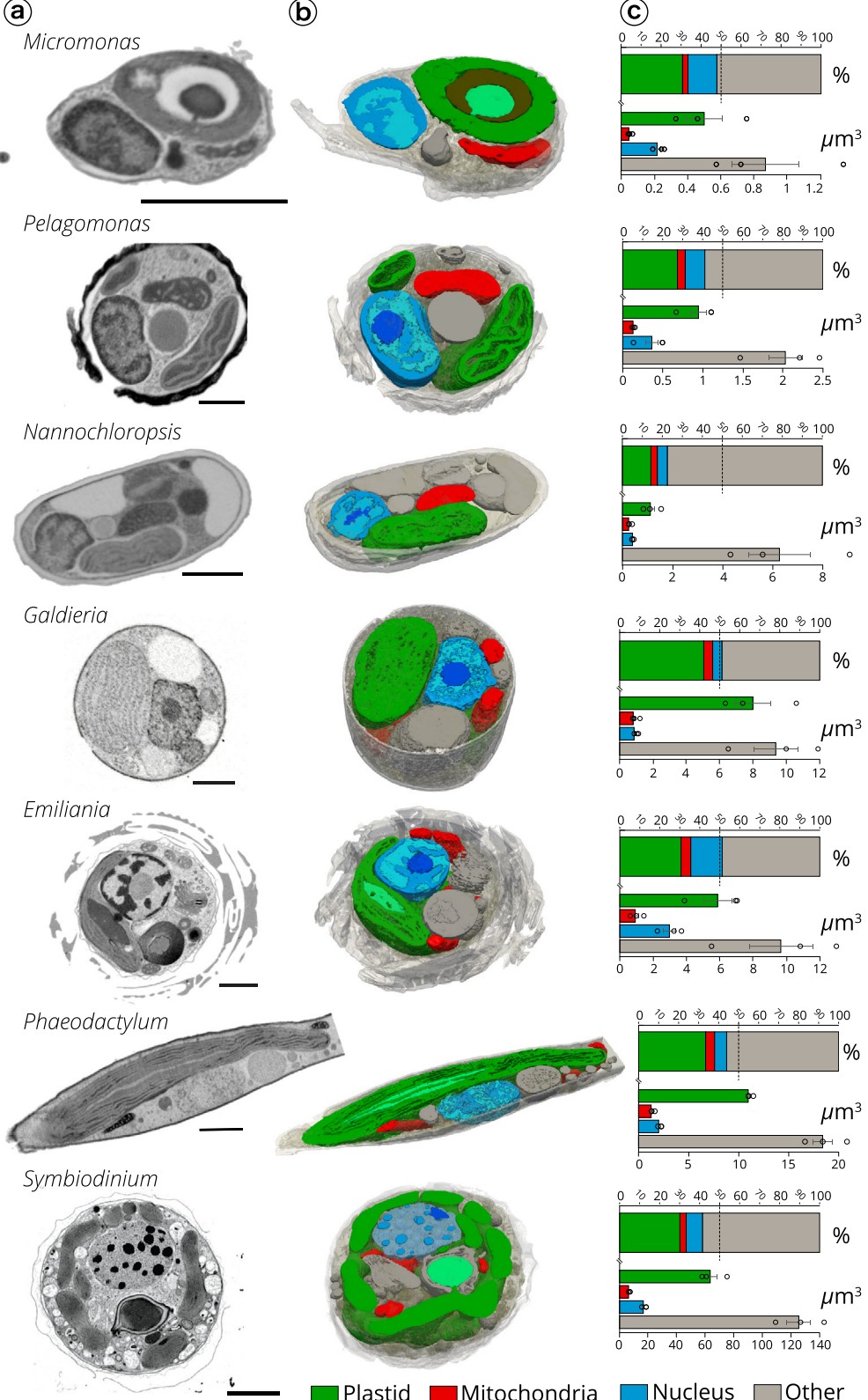

**Fig. 2 Internal cell architecture of phytoplankton cells. a** Sections through cellular 3D volumes, segmented from FIB-SEM images of whole cells of *Micromonas* (stack of frames in Supplementary Movie 1), *Pelagomonas* (Supplementary Movie 2), *Nannochloropsis* (Supplementary Movie 3), *Galdieria* (Supplementary Movie 4), *Emiliania* (Supplementary Movie 5), *Phaeodactylum* (Supplementary Movie 6) and *Symbiodinium* (Supplementary Movie 7). Sections are representatives micrographs of an experiment repeated three times with similar results Scale bar: 1 μm. **b** Segmentations highlight the main subcellular compartments: green: plastids (containing thylakoids and pyrenoids—light green—in some cell types); red: mitochondria; blue: nuclei (with different intensities of staining possibly corresponding to euchromatin—light blue—heterochromatin—blue and the nucleolus—dark blue); grey: other compartments. Segmentations are representatives tomograms of an experiment repeated three times with similar results. **c** Volume occupancy by the different subcellular compartments in different microalgal cells. Top plot: % of cell-volume occupation; bottom plot: absolute volume sizes. Data refer to three cells ± s.d. for every species.

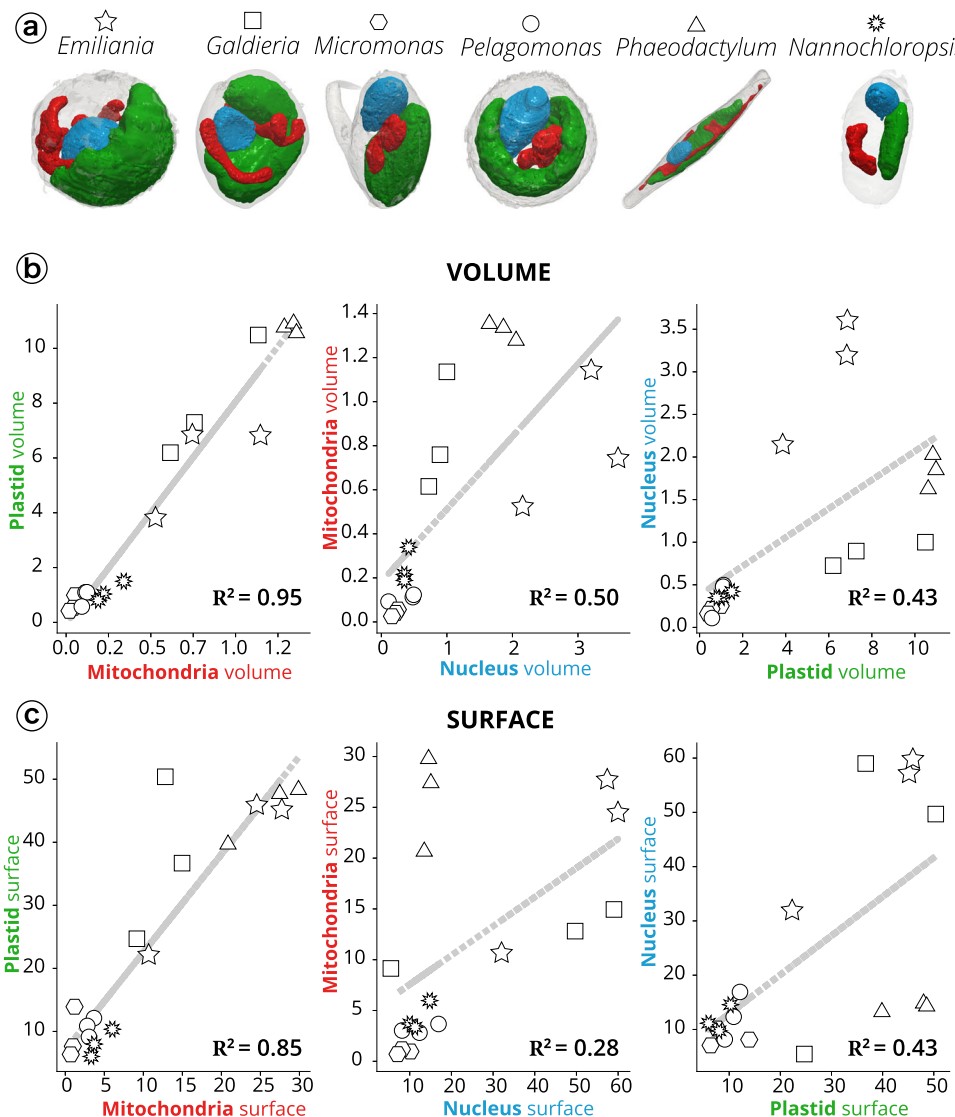

**Fig. 3 Morphometric analysis of phytoplankton members. a** 3D topology of the main organelles (green: plastids; red: mitochondria; blue: nuclei) in the different cell types. **b** Volume relationships in different subcellular compartments, as derived from quantitative analysis of microalgal 3D models. **c** Surface relationships in different subcellular compartments, as derived from quantitative analysis of microalgal 3D models. Three cells were considered for every taxum. Stars: *Emiliania*; squares: *Galdieria*; hexagons: *Micromonas*; circles: *Pelagomonas*; triangles: *Phaeodactylum*; suns: *Nannochloropsis*. *Symbiodinium* cells were not considered in this analysis, because their size, which largely exceeds the other (Supplementary Fig. 3), prevents a meaningful analysis of the volume/surface relationships.

Distances ≤90 nm may represent an 'upper limit' for contacts. Using this criterion, areas became larger in *Phaeodactylum* (15.7 ± 0.3% of the plastid surface Fig. 4b), and evident in all the tested organisms. However, due to the quite large intermembrane distance, areas calculated with this criterion likely represent a proximity between plastids and mitochondria, rather than genuine contact sites between the two organelles mediated by protein machineries, as observed in the case of other organelle-organelle interactions[58–60].

**Subcellular features of energy-managing organelles**. Besides providing information on the topology of organelles, our 3D images had enough resolution to explore sub-organelle features. We exploited this possibility to investigate the possible conservation of structural architectures within plastids and mitochondria (Fig. 5 and Supplementary Fig. 4), seeking for signatures of structural constraints related to cellular energy management. Plastids were mostly occupied by thylakoid membranes and the

stroma, and by the carbon-fixing pyrenoid (Fig. 5a), a Rubisco-rich matrix that was absent in *Pelagomonas*[46], *Galdieria*[61] and *Nannochloropsis*[62].

In two taxa (*Phaeodactylum* and *Emiliania*), we observed thylakoids crossing the pyrenoid matrix (Fig. 5a). These pyrenoid membranes (also called pyrenoid tubules in *Chlamydomonas*[6]) displayed different topologies: we observed parallel stacks in the diatom (Supplementary Fig. 4) and a more branched structure in *Emiliania*, reminiscent of that recently reported in *Chlamydomonas*[6,63]. *Micromonas* and *Symbiodinium* contained thylakoid-free pyrenoids that were almost completely surrounded by starch sheaths (Fig. 5a). Few stalks ensure the connection between pyrenoid and the plastid, possibly to facilitate the diffusion of Rubisco substrates and products as previously proposed[6,64,65], see also the review[66]. Unlike *Micromonas*, the pyrenoid of the dinoflagellate *Symbiodinium* was not centred in the plastid, but instead protruded towards the cytosol, being surrounded by a shell of cytosolic rather than stromal starch[66–68].

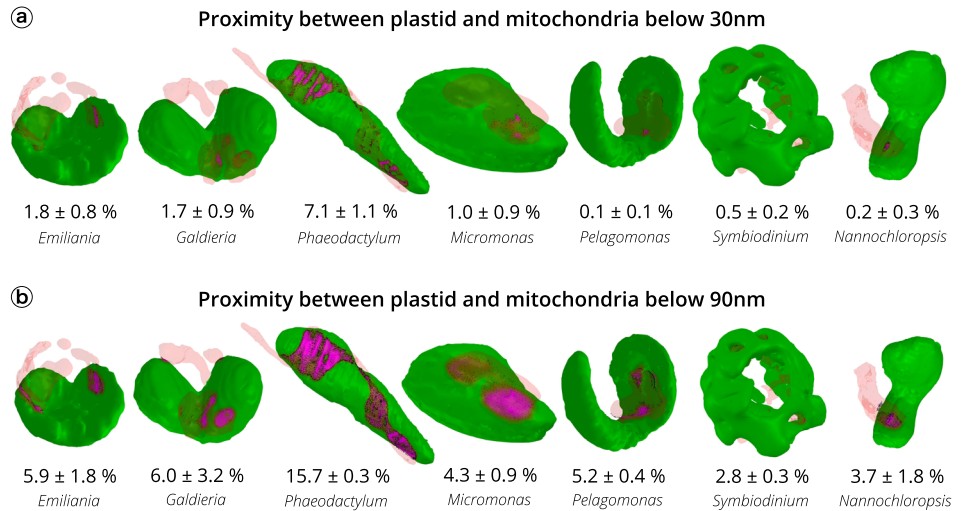

**Fig. 4 Proximity between plastids and mitochondria in different phytoplankton members.** Green: plastid surface. Red: mitochondria surface. Magenta: proximity surface (i.e. points at a distance ≤30 nm (panel **a**) or ≤90 nm (panel **b**) between mitochondria and plastids. Data refer to three cells ± s.d. for every species.

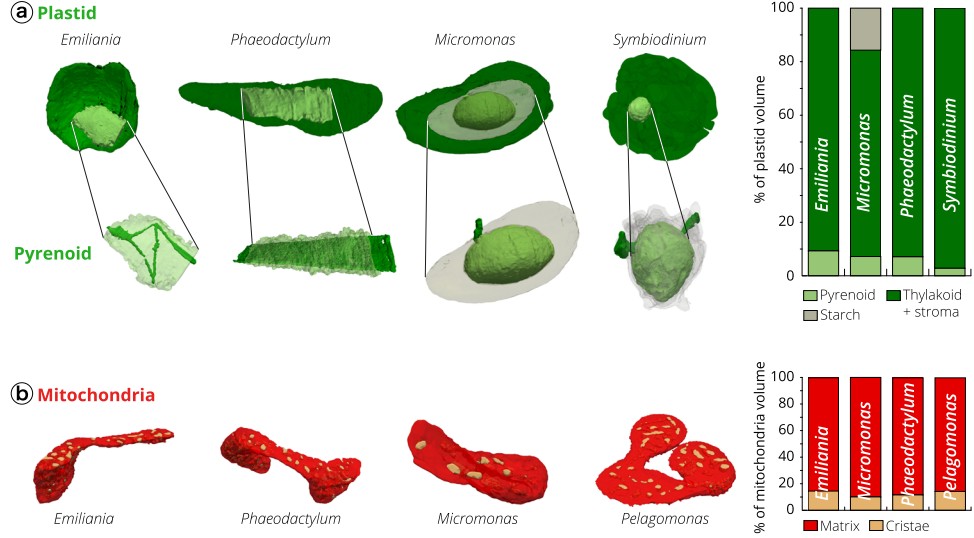

**Fig. 5 Architecture of the mitochondria and plastids of different phytoplankton taxa. a** Topology of the plastid. Whole plastid images and focus on the $CO_2$-fixing compartment (pyrenoid) topology in *Emiliania*, *Phaeodactylum*, *Micromonas* and *Symbiodinium* cells. The 3D reconstruction displays the thylakoid network (dark green) crossing the pyrenoid matrix (light green). If present (*Micromonas* and *Symbiodinium*), a starch layer surrounding the pyrenoid is shown in grey. The histogram recapitulates volume occupancy by sub-plastidial structures (thylakoids, matrix, starch, pyrenoid). Note that starch is cytosolic in *Symbiodinium*, and therefore its volume is not considered in the graph. **b** Topology of mitochondrial compartments. Red: mitochondrial matrix; yellow: cristae. The histogram recapitulates volume occupancy by mitochondrial subcompartments (in the matrix and within the cristae). See Supplementary Fig. 4 for sections through plastids and mitochondria.

Despite the differences in the pyrenoid topology, the ratio of pyrenoid/plastid volumes was preserved in three out of the four microalgae lineages where this compartment was present (7.1 ± 1.2%, 9.3 ± 1.4%, 7.2 ± 1.2% for *Phaeodactylum*, *Emiliania*, *Micromonas*, respectively, Fig. 5a and Supplementary Dataset 1). This constant ratio highlights the importance of maintaining a proper balance between the subcompartments performing light harvesting (the photosynthetic membranes) and $CO_2$ fixation (the pyrenoid). An exception to this observation is *Symbiodinium*, where the pyrenoid occupies a much lower fraction of the plastid volume (2.8 ± 0.2%). Our quantitative morphometric analysis provides a possible rationale for this difference. We found that the pyrenoid surface/volume ratio (an important parameter for gas exchange in this compartment, and therefore for $CO_2$

assimilation) is 3–5 time higher in *Phaeodactylum*, *Emiliania*, *Micromonas* (20.6 ± 6, 12.3 ± 2.6 and 15.1 ± 2.4, respectively) than in the dinoflagellate (4.7 ± 2.3). A much lower surface to volume ratio may represent a functional constraint for carbon assimilation. Therefore, we propose that the large increase in the plastid volume of *Symbiodinium* (63.5 ± 9.5 μm³ when compared to 11.0 ± 0.3 μm³, 5.9 ± 1.8 μm³ and 0.5 ± 0.2 μm³, in *Phaeodactylum*, *Emiliania* and *Micromonas* respectively, see also Supplementary Data 1) cannot be followed by a commensurate expansion of the pyrenoid volume (1.8 ± 0.3 μm³ vs 0. 8 ± 0.1, 0.5 ± 0.2 and 0.05 ± 0.03 μm³, respectively), to avoid an excessive decrease of the pyrenoid surface/volume ratio in this alga.

Overall, our volumetric analysis of the pyrenoid suggests that both the surface to volume ratio and the volumetric ratio between

the plastid and the pyrenoid are important parameters for the photosynthetic metabolism. This concept of constant volumetric ratios within energy-producing organelles is corroborated by our analysis of mitochondria. In these organelles, we found that the ratio between the volume of the cristae and the matrix (Fig. 5b) is relatively constant in these cells (11.6 ± 2.8%, 14.2 ± 2.6%, 14.5 ± 2.9%, 10.1 ± 5.9% in *Phaeodactylum*, *Pelagomonas*, *Emiliania* and *Micromonas*, respectively), despite differences in the shape (Fig. 3a and Supplementary Fig. 2) and overall volumes of their mitochondria (Fig. 2c).

**Microalgal subcellular architectures and physiological responses.** The finding that plastid-mitochondria interactions and sub-organelle volume partitioning are relatively well conserved features of phytoplankton suggests that these subcellular features could have been evolutionarily-selected to ensure proper microalgal fitness. To test this hypothesis, we looked at possible modifications in the above-mentioned parameters upon exposing microalgae of a given species to changing environmental conditions. For these experiments, we concentrated on laboratory model algae (*Phaeodactylum* and *Nannochloropsis*), which can easily be grown in different conditions.

We first focused on *Phaeodactylum* cells under different light intensities, i.e. a type of environmental modification that is often experienced by diatoms[69] in their natural milieu. Cells exposed to low light (LL: 40 μmol photons m$^{-2}$ s$^{-1}$) or high light (HL: 350 μmol photons m$^{-2}$ s$^{-1}$) led to modification of both respiratory and photosynthetic performances (Fig. 6), in line with previous reports[5]. Comparative analysis of 3D models of cells from LL and HL conditions (Fig. 6a) revealed substantial changes in the morphology of the cells at the level of plastid and mitochondria.

The volume occupied by mitochondria showed an almost two-fold increase in HL (from 3.9 ± 0.2 to 6.6 ± 0.7% Fig. 6b), consistent with the enhanced respiratory activity. Conversely, the overall plastid volume reduced slightly from 33.7 ± 1.8 to 24.7 ± 6.7%. This reduction (already reported in the case of *Phaeocystis antarctica*[24]) was not accompanied by changes in the pyrenoid volume (2.4 ± 0.6 vs 3.2 ± 0.9% of the cell volume) leading to an almost two-fold augmentation of the pyrenoid occupancy in the plastid (from 7.0 ± 1.3 to 13.2 ± 2.5%), at the expense of the thylakoids plus the stroma (Fig. 6b). This increase likely accounts for the augmented photosynthetic activity (from 37 ± 10 nmol O$_2$ mL$^{-1}$ min$^{-1}$ to 59 ± 5 nmol O$_2$ mL$^{-1}$ min$^{-1}$) observed between LL and HL acclimated cells (Fig. 6c). Indeed, photon capture by the thylakoids in HL no longer limits the photosynthetic flux, which is, instead, set by the turnover of the carbon assimilating enzymes. The finding that plastid-mitochondrial proximity increased between HL and LL cells (+36 ± 14% at ≤30 nm and +25 ± 7% at ≤90 nm, Fig. 6d) also provides a possible rationale for the enhanced photosynthesis observed in HL cells. Indeed, previous work showed that organelle interactions are an advantage for carbon assimilation in diatoms, to facilitate energetic interactions between the two cell engines[5].

Next, we compared the physiology and subcellular features of *Nannochloropsis* cells exposed to two trophic conditions. Previous studies highlighted the ability of this alga to metabolize external carbon sources under photosynthetic conditions (mixotrophy) to improve growth[70–72]. We reproduced the reported growth enhancement when cells were shifted from phototrophy (without

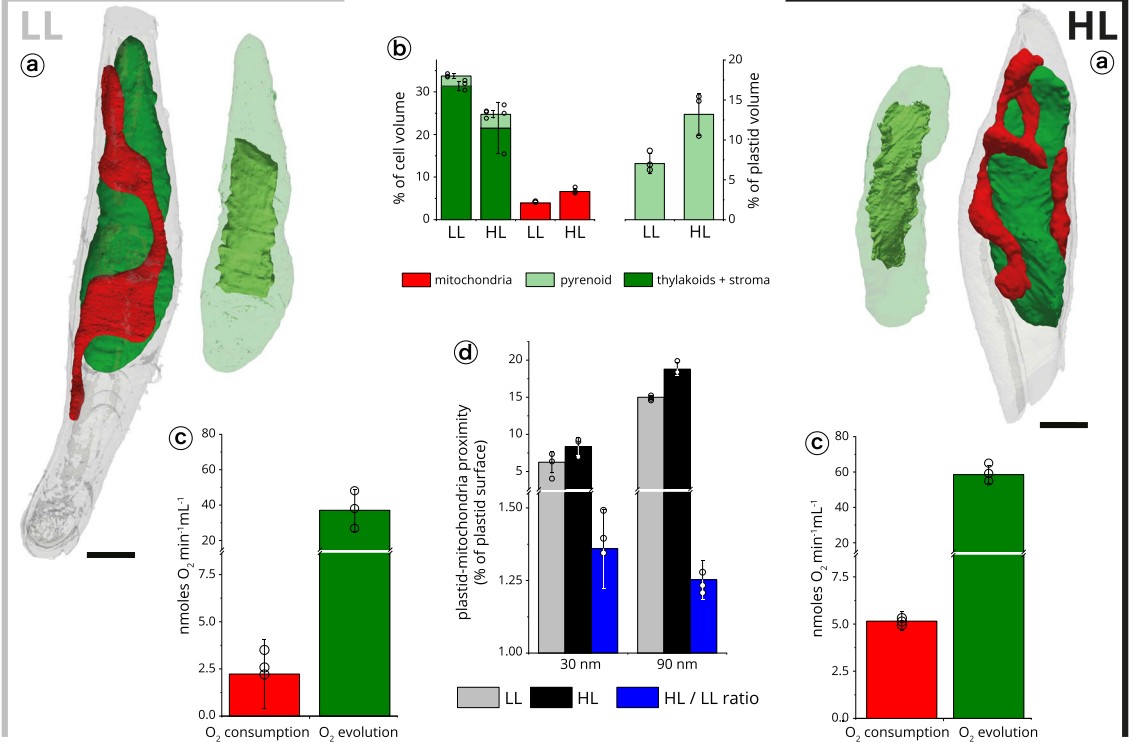

**Fig. 6 Structural analysis of light acclimation in *Phaeodactylum tricornutum*. a** Cells were imaged at two different light regimes: LL (40 μmol photons m$^{-2}$ s$^{-1}$, left) and HL (350 μmol photons m$^{-2}$ s$^{-1}$, right). Scale bar: 1 μm. **b** Volume occupancy by the plastids (dark green), mitochondria (red) and pyrenoid (light green) in the two conditions. Data refer to three cells ± s.d. for each growth condition. **c** Respiratory activities (red) and photosynthetic capacities (green) are indicated for LL (left) and HL (right) cells. Data refer to three biological samples ± s.d. for each growth condition. **d** Plastid-mitochondria proximity surface points in LL and HL cells, measured at ≤30 nm (grey) and ≤90 nm (black). At both distances, proximity areas points are increased by around 25% (blue) upon HL transition. Data refer to three cells ± s.d. for each growth condition.

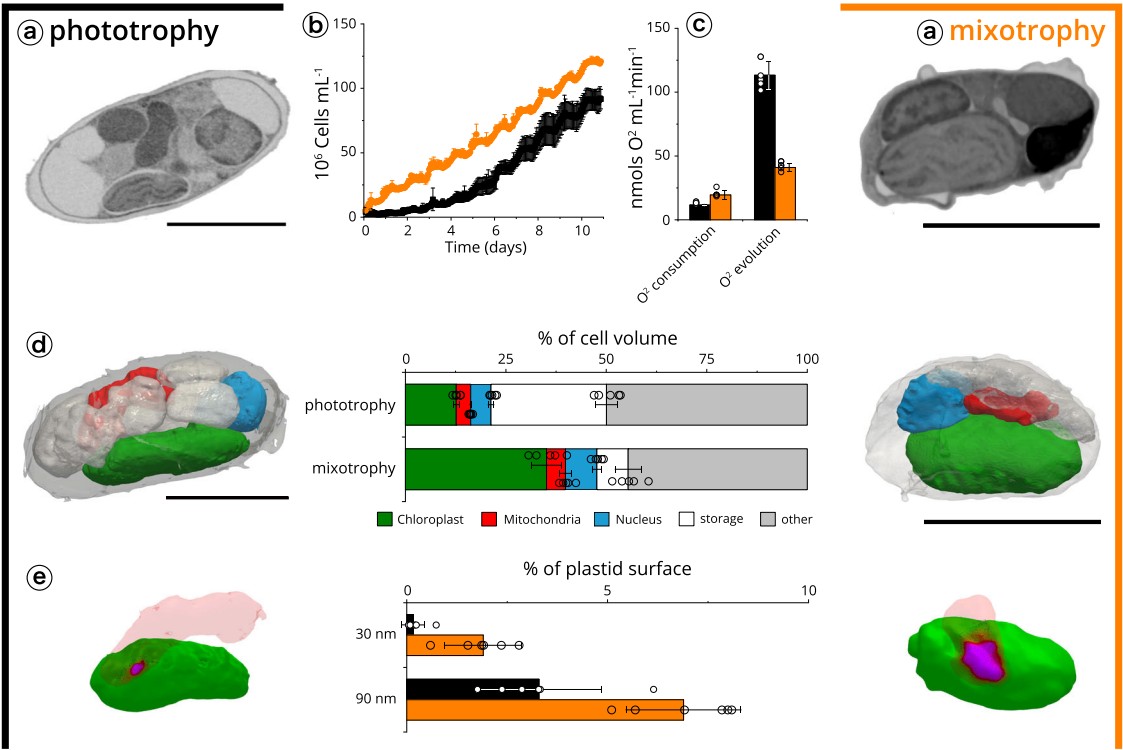

**Fig. 7 Plastid-mitochondria interactions are modified by trophic regimes in *Nannochloropsis*. a** Cells were imaged after growth in phototrophic (left) and mixotrophic (right) conditions. Sections are representatives micrographs of an experiment repeated three times with similar results. Scale bar: 2 μm. **b** Cell growth in phototrophic conditions (black) and mixotrophic conditions (orange). Data refer to three biological samples ± s.d. for each growth condition. **c** Oxygen consumption (respiration) and evolution (photosynthesis). Data refer to six biological samples ± s.d. for each growth condition. **d** Cell-volume occupancy by the different subcellular compartments in different microalgal cells. Green: plastid; red: mitochondria; blue: nuclei; white: storage vesicles; grey: other. Data refer to five cells ± s.d. for each growth condition. Scale bar: 2 μm. **e** analysis of proximity surface points (magenta) between plastid (green) and mitochondria (red). Data refer to five cells ± s.d. for each growth condition.

organic carbon) to a carbon-rich medium[73], and found that changes in the trophic lifestyle of this alga were also accompanied by substantial physiological and morphological changes (Fig. 7a). In particular, growing algae in mixotrophy enhanced their respiratory capacity (from $11.5 \pm 0.8$ to $18.9 \pm 2.9$ nmol $O_2$ mL$^{-1}$ min$^{-1}$, Fig. 7c) while decreasing their photosynthetic activity (from $113.2 \pm 11.1$ nmol $O_2$ mL$^{-1}$ min$^{-1}$ to $41.0 \pm 3.2$ nmol $O_2$ mL$^{-1}$ min$^{-1}$, Fig. 7c). These changes were accompanied by a modification in the cell organization. Phototrophic cells were largely filled with reserve vesicles ($28.7 \pm 2.9\%$ of the cell volume), thereby reducing the volume left to the organelles ($21.3 \pm 0.2\%$). The bulkiness of reserve vesicles was reduced in mixotrophic cell ($7.8 \pm 3.4\%$), in which the volume occupied by organelles (Fig. 7d) recovered a value ($47.7 \pm 1.1\%$) similar to the one observed in other algae in phototrophic conditions (Fig. 2). This reduced storage capacity in mixotrophic cells may stem from a higher consumption of lipid reserves[73], caused by the less favourable balance between photosynthesis—which produces reserves and respiration—which consumes them.

Plastid-mitochondria proximity increased in cells acclimated to mixotrophy (Fig. 7e). The effect was substantial when calculated using an organelle distance of ≤30 nm (from $0.16 \pm 0.25$ to $1.8 \pm 0.7$) and still significant (two-fold) at ≤90 nm (from $3.3 \pm 1.5$ to $6.9 \pm 1.2$). Although the proximity surface between the organelles is small, its increase could be relevant in the frame of the observed physiological changes. Plastid-mitochondria proximity may facilitate energy exchanges between the organelles in *Nannochloropsis*, to readjust the balance between the two cell organelles according to the environmental conditions. Alternatively, proximity could mediate lipid exchange between plastids and

mitochondria[53], contributing to the structural changes observed between the two trophic conditions.

By optimizing sample preparation, image acquisition, segmentation and 3D reconstruction, for a quantitative FIB-SEM tomography workflow, we have demonstrated that 3D whole-cell models can be efficiently created, providing a unique resource to quantitatively compare cellular morphological features. Our analysis of phytoplankton pinpoints conserved features (cell-volume fractions occupied by the main organelles, plastid-mitochondria proximity) and highlights the existence of constant surface/volumetric ratios within the energy-producing organelles, exemplified by the surface to volume ratio in mitochondria and in the pyrenoid. These characteristics imply the existence of topological constraints, presumably related to organelle function at the level of energy management for carbon assimilation. Our investigations of light acclimation in *Phaeodactylum* cells and acclimation to different trophic lifestyles in *Nannochloropsis* are consistent with this hypothesis, as topological modifications in their cellular engines accompanied physiological changes. These data highlight the intimate links between cellular structures, energy balance and physiological responses. Associating the approach we have developed with cryo-electron tomography, with chemical imaging[74], or with correlative microscopic studies[75,76] will vastly improve investigations of phytoplankton as well as vascular plants, e.g. to study the impacts of climate change scenarios[77]. It will be critical to assess how changes in temperature and nutrient availability in the oceans affect the subcellular features and acclimation capacity of these key phototrophic microorganisms, so as to predict their future activity at the global scale.

## Methods

**Species**. The species used in this work (Supplementary Table 1) were chosen on the basis of their representativeness of phytoplankton taxa that are ecologically relevant or of their ability to successfully grow in variable laboratory conditions.

**Algal cultivation**. *Phaeodactylum* CCAP 1055/3 was obtained from the Culture Collection of Algae and Protozoa, Scottish Marine institute, UK. Cells were grown in artificial seawater (ESAW)[78] using ten times enriched nitrogen and phosphate sources (5.49 mM $NaNO_3$ and 0.224 mM $NaH_2PO_4$; called "10X ESAW"[79]). Cells were grown in 50 mL flasks in a growth cabinet (Minitron, Infors HT, Switzerland), at 19 °C, a light intensity of 40 μmol photon m$^{-2}$ s$^{-1}$, a 12-h light /12-h dark photoperiod (unless otherwise specified) and shaking at 100 rpm. *Galdieria* SAG21.92 was obtained from the University of Dusseldorf (Germany) and was grown in sterile 2XGS modified Allen medium, pH 2.0 (ref. [80]) at 42 °C under the same light conditions. Cells were grown in 250 mL flasks (50 mL culture volume). *Nannochloropsis* CCMP526 was also cultivated in artificial seawater (10X ESAW). Photoperiod was 12-h light /12-h dark. Cells were shifted from phototrophic to mixotrophic conditions through the addition of 5% Lysogeny Broth (LB) to the growth medium. *Micromonas* RCC 827, *Pelagomonas* RCC 100, *Emiliania* RCC 909 (grown in K medium at 20 °C), and *Symbiodinium* RCC 4014 (grown in *f*/2 medium at 20 °C) were obtained from the Roscoff Culture Collection (http://www.roscoff-culture-collection.org/)[81] and maintained at a light intensity of 60–80 μmol photons m$^{-2}$ s$^{-1}$, in a 12-h light /12-h dark photoperiod, without shaking.

*Nannochloropsis* growth was measured following changes in the culture optical density at 650 nm. Changes were calibrated with cell numbers in both control and mixotrophic cultures.

**Photophysiology measurements**. Oxygen exchanges were measured with a Clark-type electrode (Hansatech Instruments, UK) at 20 °C, with respiration and gross photosynthesis quantified by measuring the slope in the dark and upon light exposure (intensity 300 μmol photons m$^{-2}$ s$^{-1}$), respectively.

The parameter Fv/Fm (maximum yield of photosystem II photochemistry)[82] was used to compare the photosynthetic capacity of the tested strain with earlier data in the literature, as a proxy for their physiological state. $F_v/F_m$ was measured with a Speedzen 3 chlorophyll fluorescence imaging setup (Biologic, France). It was calculated as $(F_m-F_0)/F_m$, where $F_0$ is the minimum fluorescence yield, determined at open photosystem II reactions centres and $F_m$ is the maximum fluorescence yield, measured upon closing the photosystem II reaction centres with a short (150 ms) saturating light pulse.

**Sample preparation for electron microscopy**. Sample preparation protocols were adapted from reference[41] to optimize the contrast for 3D electron microscopy imaging and therefore facilitate image segmentation through pixel classification. Live cells were cryofixed using high-pressure freezing (EM HPM100, Leica, Germany) in which cells were subjected to a pressure of 210 MPa at −196 °C, followed by freeze substitution (EM ASF2, Leica, Germany). Prior to cryo fixation, the microalgal cultures were concentrated by gentle centrifugation for 10 min (1000 *g*). For the freeze substitution, a mixture 2% (w/v) osmium tetroxide and 0.5% (w/v) uranyl acetate in dried acetone was used. The freeze-substitution machine was programmed as follows: 60–80 h at −90 °C, heating rate of 2 °C h$^{-1}$ to −60 °C (15 h), 10–12 h at −60 °C, heating rate of 2 °C h$^{-1}$ to −30 °C (15 h), and 10–12 h at −30 °C, quickly heated to 0 °C for 1 h to enhance the staining efficiency of osmium tetroxide and uranyl acetate and then back to −30 °C. The cells were then washed four times in anhydrous acetone for 15 min each at −30 °C and gradually embedded in anhydrous araldite resin. A graded resin/acetone (v/v) series was used (30, 50 and 70% resin) with each step lasting 2 h at increased temperature: 30% resin/acetone bath from −30 °C to −10 °C, 50% resin/acetone bath from −10 °C to 10 °C, 70% resin/acetone bath from 10 °C to 20 °C. Samples were then placed in 100% resin for 8–10 h and in 100% resin with the accelerator BDMA for 8 h at room temperature. Resin polymerization finally occurred at 65 °C for 48 h.

**FIB-SEM acquisition imaging**. Focused ion beam (FIB) tomography was performed with either a Zeiss NVision 40 or a Zeiss CrossBeam 550 microscope (Zeiss, Germany), both equipped with Fibics Atlas 3D software for tomography (Supplementary Fig. 1a). The resin block containing the cells was fixed on a stub with carbon paste, and surface-abraded with a diamond knife in a microtome to obtain a perfectly flat and clean surface. The entire sample was metallized with 4 nm of platinum to avoid charging during the observations. Inside the FIB-SEM, a second platinum layer (1–2 μm) was deposited locally on the analysed area to mitigate possible curtaining artefacts. The sample was then abraded slice by slice with the Ga$^+$ ion beam (generally with a current of 700 nA at 30 kV). Each freshly exposed surface was imaged by scanning electron microscopy (SEM) at 1.5 kV and with a current of ~1 nA using the in-lens EsB backscatter detector. For algae, we generally used the simultaneous milling and imaging mode for better stability, and with an hourly automatic correction of focus and astigmatism. For each slice, a thickness of 8 nm was removed, and the SEM images were recorded with a pixel size of 8 nm, providing an isotropic voxel size of 8 × 8 × 8 nm$^3$. Whole volumes were imaged with 800–1000 frames, depending on the species. Due to its reduced cell

dimensions, the voxel size was reduced to 4 × 4 × 4 nm$^3$ in the case of *Micromonas*, resulting in higher resolution datasets with approximately 350–500 frames/cell.

**Image processing**. As a first step of image processing, regions of interest (ROIs) containing cells were cropped from the full image stack. This was followed by image registration (stack alignment), noise reduction, semi-automatic segmentation of the ROIs, 3D reconstruction of microalgae cells and morphometric analysis. Several problems may be encountered during these steps. Raw stacks consist of big data (50–100 GB for the whole imaged volume, containing several cells) that do not necessarily fit into the computer main memory (RAM). Moreover, cryo-substituted cells generate less contrasted images than cells prepared with chemical fixation. Therefore, the first step in building a robust 3D model consists in 'isolating' a given ROI (e.g. an organelle) from other compartments, to obtain a smaller stack size that can be easily worked with (in practice, we worked with substacks that were around 10% of the original stack size).

Single cells were isolated by cropping in three dimensions using the open software Fiji (https://imagej.net/Fiji, Supplementary Fig. 1a). Image misalignment was corrected using the template matching ("align slices in stack") option implemented in Fiji. This function finds the most similar image pattern in every slice and translates them to align the landmark pattern across the stack (https://sites.google.com/site/qingzongtseng/template-matching-ij-plugin) (Supplementary Movies 1-9). Aligned image stacks were filtered to remove noise using Python[83] and OpenCV (https://opencv.org). Filtering techniques were chosen to highlight contours while removing noise in the images. Depending on the species, we found that the osmium staining was not homogeneously distributed. Therefore, it was not possible to filter raw datasets of every species with the same method. Based on the effectiveness in highlighting organelle boundaries, different filters were used for the different microalgae (Supplementary Fig. 1b). Application of a linear Gaussian filter followed by sharpening to remove noise and enhance contours, which is widely used and easy to implement[84], was used to process raw datasets of *Emiliania*, *Micromonas*, *Phaeodactylum* and *Pelagomonas*. However, this method was less effective when applied to raw datasets of *Galdieria* and *Symbiodinium*, where using the median filter proved to be a better de-noising option. These choices reflect the different cellular features and biochemical composition of each taxon (e.g. the presence of a thick cell wall in *Galdieria*), which results in variable contrast.

**Segmentation**. Segmentation of organelles, vesicular networks, vacuoles and storage compartments was carried out with 3D Slicer software[85] (www.slicer.org, Supplementary Fig. 1c), using a manually-curated, semi-automatic pixel clustering mode (3 to 10 slices are segmented simultaneously for a given ROI). We assigned colours to the ROIs using paint tools and adjusted the threshold range for image intensity values. The ROIs were annotated and the corresponding label map was run into the model maker module from 3D slicer (Supplementary Fig. 1c), to generate corresponding 3D models that were exported in different formats (.stl, obj, vtk, ply, mtl). For further analysis, we used the.stl mesh, which proved to be most suitable for 3D analysis in our workflow (Supplementary Table 2).

**3D reconstructed model**. A 3D filtering process was needed to refine the model and reduce the size of the file (see Supplementary Table 2). In our case, 3D models generated by 3D Slicer were imported into the open source software MeshLab[86,87] which automatically removed some 'isolated islands'. Models were further edited manually within MeshLab to eliminate remaining isolated islands erroneously annotated as ROIs. We also performed a remeshing process to facilitate 3D modelling, visualization and animation. Using MeshLab, we simplified meshes ('mesh decimation', Supplementary Fig. 1d) to reduce the model nodes and faces down to 25% of the original data without modifying morphometric values, such as surfaces and volumes (Supplementary Table 2). Every 3D model was imported into Paraview[88] (Supplementary Fig. 1d) to visualize 3D objects and understand their relationships. Blender (www.blender.org) was used for object animation (Supplementary Movie 10).

**Morphometric evaluations**. Measurement of volumes, surface area, and the minimum distance between meshes) were performed using Numpy-STL (https://pypi.org/project/numpy-stl/) and TRIMESH (https://trimsh.org/trimesh.html) packages of Python (Supplementary Table 3). This Python package is faster than MeshLab, with obvious advantages in terms of analysis of large files (>500 MB).

**Surface and volume measurements**. Surfaces and volumes were computed using the discrete mesh geometry, with surfaces computed directly from mesh triangles and volumes computed from the signed volume of individual tetrahedrons, assuming a closed surface (watertight mesh, Supplementary Fig. 1e). Briefly, to compute the surface, we iterated over all the triangles of the mesh. The computation of the cross product between two edges of a given triangle gives a vector whose magnitude is twice the area of said triangle. Then, the sum of all these areas provides the total surface area of the mesh. We then computed the signed volume of all tetrahedrons, which goes from the origin (0,0,0) to each triangle present in the mesh. Assuming a closed surface (watertight mesh), summing all those volumes gives the volume of the mesh[89]. A simple implementation of those algorithms is provided in Supplementary dataset 2.

**Distance between organelles**. Using the Trimesh Python module, the minimal distance between two meshes was calculated based on the closest points between two triangular meshes. Hence, the surface proximity areas were quantified based on: (i) calculating the minimal distance between each vertex of the plastid mesh to the mitochondria mesh (for 3 cells of every species), and then by (ii) gathering mesh vertices according to a given distance threshold to generate proximity surfaces. Two distance thresholds were chosen for this analysis: ≤30 nm, the 'average' one, to define possible contact points[54–56] and ≤90 nm, defining the 'upper limit' for organelle proximity[57,90]. The corresponding surfaces were then compared to the total plastid surface (Supplementary dataset 3).

**Reporting summary**. Further information on research design is available in the Nature Research Reporting Summary linked to this article.

## Data availability
The authors declare that all the data supporting the findings of this study are available within the paper and in its supplementary information files. Raw FIB-SEM stacks are available at https://www.ebi.ac.uk/biostudies/studies/S-BSST575. Source data are provided with this paper.

## Code availability
The computer codes supporting the findings of this study are available in the supplementary data 2 and supplementary data 3. The proximity distance computation code is available at: https://gitlab.com/clariaddy/mindist. The metrics computation code is available at: https://gitlab.com/clariaddy/stl_statistics.

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

## Acknowledgements

The authors thank the Roscoff Culture Collection that provided phytoplankton strains and Noan Le Bescot (Ternog Design) for help in the conception and realization of the figures of this article. This project received funding from the European Research Council: ERC Chloro-mito (grant no. 833184) to G.F., D.F., G.C. Research was also supported by a Défi X-Life grant from CNRS to J.D., funds from the CEA DRF impulsion FIB-Bio program to J.D., P.-H.J., B.G., C.M., D.F., G.S.; the European Union's Horizon 2020 research and innovation programme CORBEL under the grant agreement No 654248 to J.D., the LabEx GRAL (ANR-10-LABX-49-01), financed within the University Grenoble Alpes graduate school (Ecoles Universitaires de Recherche) CBH-EUR-GS (ANR-17-EURE-0003) to C.W., J.D., P.-H.J., B.G., C.M., F.C., G.C., G.S., D.F., G.F. and the ANR 'Momix' (Projet-ANR-17-CE05-0029) to G.C., G.F. This work used the platforms of the Grenoble Instruct-ERIC centre (ISBG; UMS 3518 CNRS-CEA-UGA-EMBL) within the Grenoble Partnership for Structural Biology (PSB), supported by FRISBI (ANR-10-INBS-05-02) and GRAL to B.G., C.M., G.S. The electron microscope facility is supported by the Auvergne-Rhône-Alpes Region, the Fondation Recherche Medicale (FRM), the funds FEDER and the GIS-Infrastructures en Biologie Santé et Agronomie (IBiSA) to B.G., C.M., G.S.; J.D. was supported by ATIP-Avenir program. C.U. is supported by a joint UGA-ETH Zurich PhD grant (to G.F. and S.C.Z.) in the framework of the "Investissements d'avenir" programme (ANR-15-IDEX-02).

## Author contributions

C.U. designed the work, performed image treatment; made scripts and drafted the manuscript; J.D. designed the manuscript, prepared samples for FIB-SEM, interpreted data and drafted the manuscript; P.-H. J. conceived the work, performed FIB-SEM imaging and drafted the manuscript; S.F. characterized the physiological responses of LL and HL *Phaeodactylum* cells; B.G. performed sample preparation; J.B.K. performed image treatment and made scripts; D.D.B. characterized the physiological responses of phototrophic and mixotrophic *Nannochloropsis* cells; C.M. performed cryo fixation. G.A. optimized growth of phototrophic and mixotrophic *Nannochloropsis* cells; F. Chevalier. optimized samples growth; C.S. characterized the physiological responses of LL and HL *Phaeodactylum* cells; N.L.S. performed FIB-SEM imaging; R.T. performed FIB-SEM

imaging; F. Courtois. interpreted data; G.C. optimized samples growth and interpreted data; Y.S. performed FIB-SEM imaging and participated to manuscript drafting; G.S. designed the work and participated to manuscript drafting; S.C.Z. designed the work interpreted data and drafted the manuscript; D.F. designed the work, prepared samples for FIB-SEM, interpreted data and drafted the manuscript; G.F. designed the work; interpreted data and drafted the manuscript. All authors revised and approved the text.

## Competing interests

The authors declare no competing interests.
