## [Peer Review File · Nature Communications]

Reviewers' comments:

Reviewer #1 (Remarks to the Author):

In this manuscript, Uwizeye et al. use FIB-SEM imaging of resin-embedded cells to perform a survey of organelle morphology in six diverse species of phytoplankton. This work is a natural extension from this research group's previous FIB-SEM study of diatoms, which characterized physical interaction surfaces between mitochondria and the chloroplast (Flori et al., 2017; Nat Comm). 3D segmentation of the tomograms (3 cells examined per species) enabled quantification of the volumes occupied by various organelles (nucleus, mitochondria, chloroplast) and certain sub-organellar regions (pyrenoid, cristae, nucleolus). The cell biology and physiology of these phytoplankton species have not been well characterized before, so I think such descriptions are very important, both from an evolutionary perspective and also because these marine algae are major players in the global carbon cycle. As the Earth's climate rapidly warms, it is crucial to understand the cellular physiology of marine algae and how these algae respond to changing environmental conditions.

I am enthusiastic about this study's topic and FIB-SEM imaging approach. I enjoyed looking at the 3D cellular architecture revealed in the tomograms, which I feel provides a great starting point for future studies. However, I am disappointed that the current study does not go further to investigate the physiology of these cells. The analysis is limited to basic correlations of organelle volumes and surface areas between species, but does not begin to address the important question of how organelle architecture is remodeled in response to environmental change. If the authors were able to add some analysis addressing this topic, I would find this study very compelling.

Major Points:

1) This paper lacks a strong biological focus. The authors tease in their abstract that their approach "opens new perspectives to study acclimation responses to abiotic and biotic factors at a relevant biological scale", but they perform no experiments along these lines. I very much appreciate the long time it takes to perform the FIB-SEM analysis. However, I think the impact of this paper would be greatly increased if the authors explored at least one biological question in some detail. Some possibilities include changes in pyrenoid morphology in low vs. high CO₂ or changes in thylakoid-mitochondria contacts in different environmental conditions.

2) Part 1 of the results section (including Fig. 1) presents a detailed technical description of how the tomograms were segmented and the volumes / surface areas were measured. This part of the manuscript reads very much like a methods section, and thus should be moved to the methods. Fig. 1 should be a supplemental figure.

3) While I appreciate that this time-intensive technique limits the available statistics, the authors should be careful when making general conclusions from a small dataset (3 cells were imaged per species). For example, the authors state several times that the variable mitochondria architecture reflects the organelle's dynamic nature. Inferring dynamics requires either live-cell microscopy or at least far more static snapshots than were performed here, ideally with multiple physiological conditions. If it isn't feasible to increase the sample size, the authors should be cautious with their generalizations.

4) Besides *Phaeodactylum*, all the other algae species showed <2% of the plastid in contact with mitochondria. The authors conclude this suggests "that the whole process could be dynamic throughout the life cycle and environmental conditions, possibly being mediated by molecular actors". In the absence of experimental evidence looking at these conditions, this general conclusion feels far too speculative. How do the authors know that this small contacting surface area represents a specific interaction mediated by molecular actors? To this reviewer, a very

plausible explanation could be that these are random contacts that do not transfer energy between the organelles. If anything, the major contrast to diatoms seems to suggest this could be the case. But in the absence of additional data, this too is speculation.

5) The authors devote a large portion of the text to the pyrenoid. I agree that this is a fascinating and important topic that can be illuminated by FIB-SEM imaging. The pyrenoid has evolved convergently in each of these different species, so it is of high interest to carefully detail the variable morphology of these compartments. Unfortunately, it was hard for me to appreciate the pyrenoid architecture in the small segmentations provided in Fig. 6A. Perhaps the authors could create an additional figure showing the pyrenoid of each species in more detail, comparing slices through the raw tomograms with corresponding slices through the segmentations.

Minor Points:

Page 4: "we generate accurate 3D reconstructions of whole cells in their close-to-native states".
-- Even with the improved sample preservation provided by freeze substitution, the heavy metal staining and resin-embedding of cells performed in this study does not reveal cells in their "close-to-native state". This term should be reserved for hydrated frozen cells (e.g., cryo-EM).

Page 8: "photosynthetic membranes (thylakoids) were organised in layers of a few stacks, without the typical differentiation in grana and stromal lamellae observed in vascular plants and green algae"

-- Green algae do not have the canonical grana and stroma lamellae of plants. This should be reworded to say something like "without the subdivision into stacked and unstacked membrane domains observed in green algae".

Page 9: In Fig. 3C, the values of the normalized and absolute volumes do not match. For example, the plastid is the largest absolute volume in all species, but the cytosol is shown to occupy the largest percentage of the volume. Perhaps the color code was mixed up between the graphs (green switched with grey, red switched with blue)?

Page 10: "Altogether, the organelles (nuclei, plastids and mitochondria) filled a relatively constant fraction (40 to 55 %) of the total cell volume"

-- The Golgi and ER are also universally considered to be organelles. The authors say that the Golgi has "high variability in terms of volume occupancy" but then conclude generally that: "We interpret this conservation of organelle volumes and the variability of the other compartments as the signature of evolutionary constraints that preserve essential cellular functions". This should be reworded. The Golgi is an organelle, not an "other compartment". Specifically, the volume of the nucleus, mitochondria, and chloroplast appear to be relatively conserved between the studied species, but this is not true for all organelles.

Page 10: "...unveiled the existence of a tight correlation between plastids and mitochondria in terms of their volume. This conserved topological feature corroborates the previous molecular and physiological evidence in diatoms for energetic interactions between plastids and mitochondria."

-- It is not clear to me how a constant volume ratio between chloroplasts and mitochondria in several species corroborates energy exchange between these organelles.

Page 13: "Different levels of DNA condensation are visible. DNA is present in the form of compact chromosomes in Symbiodinium, possibly leaving a fraction of the nucleoplasm without chromatin (grey)"

-- How do the authors know that variable staining of the nucleoplasm with osmium tetroxide and uranyl acetate reflects degrees of DNA condensation? The resolution is far too limited to directly see chromatin compaction (i.e., nucleosomes). The authors should clearly state that this is an assumption they are making based on an indirect measurement (affinity for heavy metal stain),

and they should cite literature that supports this assumption. The statement that some of the Symbiodinium nucleoplasm may lack chromatin is too speculative based on the resolution of this FIB-SEM data.

Page 14: "The plastid volumes were mainly occupied by thylakoid membranes and, when present, by the pyrenoid"

-- When was the pyrenoid not observed? Do *Galdieria* and *Pelagomonas* lack pyrenoids? Is this expected based on previous literature or their phylogeny? It would be appropriate to have some discussion of why pyrenoids were absent. Is it the species, growth condition, or a technical limitation preventing detection?

Page 17: "we foresee in the future a wide utilisation of this technique to analyse subcellular and even sub-organellar structures, to help understanding physiological responses at the macromolecular level."

-- This should be reworded. FIB-SEM does not provide information at the macromolecular level.

-- The number of replicates and statistical methods used should be described in every figure legend.

-- The movie with the 3D tour of the coccolithophore segmentation is fantastic! I know it is labor-intensive, but I would love to see similar movies of the other algae.

Formatting and Edits:

Page 2: "its diversity" -> their diversity

Page 3: "A high-throughput confocal" -> High-throughput confocal

Page 3: "However, optical microscopy studies have insufficient resolution to reveal microstructural features, emphasising the need to develop complementary imaging approaches to study the cellular and subcellular bases of phytoplankton physiology and cell biology." -> However, optical microscopy studies have insufficient resolution to reveal cellular ultrastructure, emphasizing the need to develop complementary imaging approaches to study the structural basis of phytoplankton physiology.

Page 3: "3D electron microscopy (EM) imaging" -> 3D electron microscopy (EM)

Page 3: "to access the cellular and subcellular features of phytoplankton at the nanometric scale" -> to access the subcellular features of phytoplankton at the nanometer scale

Page 3: "providing contextual 3D images of whole cells at a specific time." -> imaging 3D volumes of whole cells.

Page 5: "with a workable (~1000-1200) number of frames." -> with ~1000-1200 frames.

Page 6 (Fig. 1 legend): "surface is obtained" -> surface area is obtained

Page 7: "e.g. coccolithophores in *Emiliana*" -> e.g. coccoliths in *Emiliana* (coccolithophores are the organisms, not the plate structures)

Page 8 (Fig. 2 legend): "External feature" -> External features

Page 9 (Fig. 3 legend): "cells as revealed by FIB-SEM" -> cells revealed by FIB-SEM

Page 9 (Fig. 3 legend): "Sections of cellular 3D volumes, based on FIB-SEM imaging" -> Sections through cellular 3D volumes, segmented from FIB-SEM images (or you could say tomograms)

Page 9 (Fig. 3 legend): "Images highlight" -> Segmentations highlight

Page 11 (Fig. 4 legend): "exegons" -> "Hexagons"

Page 11 (Fig. 4 legend): "prevents a correct analysis" -> prevents a meaningful correlation

Page 12: "approximately 5% of the total surface" -> approximately 7% of the total surface (according to Fig. 5)

Page 13: "explore the sub-organelles features" -> explore the sub-organelle features

Page 15: "maintaining a proper balance between the subcompartments producing light-dependent energy (the photosynthetic membranes) and the light-independent one (the CO₂ fixing compartment)." -> maintaining a proper balance between the subcompartments performing light harvesting (the photosynthetic membranes) and CO₂ fixation (the pyrenoid).

Page 16: "become more structured insofar as the vesicles detached" -> "become more structured as the vesicles detach"

Page 17: "Since this technique can be applied to all type of cells" -> Since FIB-SEM can be applied to all types of cells

Reviewed by Benjamin Engel and Wojciech Wietrzynski

Reviewer #2 (Remarks to the Author):

The manuscript "In-cell quantitative structural imaging of phytoplankton using 3D electron microscopy" by Uwizeye et al. describes the cellular and subcellular architecture of six eukaryotic microalgae based on 3D-imaging data that was obtained by focused Ion Beam Scanning EM (FIB-SEM). From the imaging data, 3D models of cells and subcellular structures were generated and compared using three-dimensional (3D) morphometric analyses. The latter led to the hypothesis that the volumetric ratio of plastids and mitochondria may be constant in the different algae lineages.

Although we lack 3D ultrastructural data for most of the here investigated algae and I find it of great importance to study ultrastructure of eukaryotic algae to better understand their biology, I have substantial concerns with respect to the developed workflow and the presentation and interpretation of the data.

In recent years, volume microscopy has become an important method in cell biology. I am aware of whole-cell 3D FIB-SEM data of a few celled photosynthetic organisms [1-5] (see below for references). To my surprise, neither the *Chlamydomonas reinhardtii* [1,2], *Emiliana huxleyi* [3], nor the dinoflagellate *Calciodinellum operosum* [4, 5] work is referenced at the appropriate position in the introduction where the authors state "This technique (FIB-SEM) has already been applied successfully to provide 3D models of eukaryotic cell" (page 3).

In contrast to the author's work, previous 3D imaging of *Emiliana* was based on cryo FIB-SEM. Cryo FIB-SEM does not involve artifact introducing sample preparation steps as does the FIB-SEM workflow used by the authors. With all due respect to the large quantity of data and the challenging work produced, the authors should know that their sample preparation protocol does

not preserve soluble calcium phases [6] and thus data sets from cells that have been prepared by the author's method lack important information. This limitation of the method and an apparent lack of knowledge about the ultrastructure of *E. huxleyi* and the process of coccolith formation led to severe misinterpretation of the cellular structures that can be seen in the *Emiliana* data set. Firstly, *Emiliana* forms coccoliths inside the cells in a membrane-bound compartment called the coccolith vesicle. Usually, one coccolith is formed at a time and always one coccolith in one coccolith vesicle [7]. Instead, the authors write: "... formation of a proccoloth (do you mean protococcoloth?) within Golgi-associated vesicles ..." (page 16). Unequivocal identification of the coccolith vesicle is only possible when it contains a nascent coccolith. Without containing a nascent coccolith the coccolith vesicle is hardly distinguishable from other membrane-bound structures. Neither in the SEM images shown in figure 7 nor the video in the supplementary material a cell with a nascent coccolith within a coccolith vesicle is visible. Therefore, I have no idea where the authors have seen the coccolith vesicle, protococcolith, and organic base plate scale for which they generated the 3D models shown in figure 7b. All I can see is that these models lack the morphology that one would expect for these structures [3, 8, 9]. The authors should provide the SEM images/image stacks, and not only 3D models. Mistakes in the identification of subcellular structures can then be corrected. The structure of which the authors claim to be the Golgi apparatus seems to me to be the reticular body [8], a complex membrane system positioned on top of the coccolith vesicle in *Emiliana*. The reticular body is in close contact with the coccolith vesicle and not the Golgi apparatus, as the authors write on page 16. The reticular body due to its large size a conspicuous organelle. The Golgi apparatus in *Emiliana* is usually smaller than the reticular body and easy to overlook. Secondly, once calcification in the coccolith vesicle is complete, the mature coccolith is secreted onto the cell surface. The coccosphere should therefore mainly contain mature coccoliths. Immature coccoliths are usually secreted to the cell surface only under stress conditions [10, 11]. The fact that the authors observe multiple immature coccoliths on the cell surface suggests that the cells experienced severe stress for several hours before they were snap frozen. Under ideal conditions it takes about 1 hour to form a coccolith [7]. The impact of stress on the volumes of organelles has not been investigated to my knowledge. I think this aspect should be clarified before comparing stressed *Emiliana* cells with cells of species that may be in a different physiological state. Thirdly, the nucleation of calcite crystals inside the coccolith vesicle takes place on the surface of a disc-shaped organic structure, which is called base plate or "organic base plate scale". The base plate scale becomes a component of the coccolith. It is usually visible inside the coccolith vesicle before the onset of calcite nucleation and at the base of nascent and mature coccoliths [12]. Therefore, the statement of the authors "... coccoliths are released outside the cells making the Organic Base Plate Scale (OBOS) easy detectable in the cell." (figure 7 legend) makes no sense to me. In none of the provided images, the coccolith base plate scale is visible to me. By the way, reference 49 is wrongly claimed to be a review of coccolithogenesis (page 16).

Another concern I have is related to the quantification of the contact areas between organelles. The authors define membranes to be in contact when their distance is less than 30 nm (page 12). Even though there is no accepted definition for contact sites, a 30 nm gap is a huge distance. Ribosomes have a diameter of ~20 nm and therefore would fit into a 30 nm gap. Ribosomes between the membranes of two organelles would speak against contact. Unfortunately, the quality and resolution of the images/videos provided is so poor that ribosomes cannot be seen. That FIB-SEM can visualize ribosomes has been shown previously (figure 5B reference [2]). In comparison to the presentation of previous FIB-SEM work in other high-ranking journals [2, 13, 14], I find the presentation of the data in this manuscript poor. Given that FIB-SEM is the only method used in this study, the data should be presented in a more detailed and precise way, making it possible for the reader to check the author's interpretation of the data. For example, the supplementary figure 1 claims to show 3D reconstructions of mitochondria of *Emiliana*, but the corresponding images to dispel doubts that this membrane system is peripheral endoplasmic reticulum are missing. The presented 3D model of *Pelagomonas calceolate* (fig 3, fig 4) is not in line with the original description of the species. The cells are described to be flagellated, elliptically shaped and to have a single chloroplast [15]. Has it been checked by sequencing of genetic marker genes that the received strain is truly *Pelagomonas calceolate*? It would not be the first time that a culture

collection has send a wrong species.

I strongly recommend adding physiological data of the cells that were subjected to imaging in future manuscript submissions to demonstrate that the cells of the different species are in a similar physiological state. In find this important for the interpretation of the volume occupancy of organelles. For example, it is well established that N-starvation induces the formation of lipids droplets in algae. Lipid droplets are therefore not a specific subcellular feature of *Phaeodactylum tricorutum* (page 7).

Finally, the workflow developed by the authors is by far not sufficiently detailed to be reproduced. Details of the filter parameter should be given and the Python script for filtering should be included to the supplementary material.

In summary, this is an interesting study, but as explained above I cannot verify the conclusions of the authors because the data is not presented in a precise and detailed way. The reported data on *Emiliana* contradict in many aspects the current knowledge on coccolith formation in this species. This misinterpretation of data makes this work not very convincing. The legend of figure 7 and last paragraph before the conclusion section should be completely revised.

References:

1. García-Cerdán JG, et al. Chloroplast Sec14-like 1 (CPSFL1) is essential for normal chloroplast development and affects carotenoid accumulation in *Chlamydomonas*. PNAS 117, 12452-12463 (2020).
2. Xu CS, et al. Enhanced FIB-SEM systems for large-volume 3D imaging. eLife 6, e25916 (2017).
3. Sviben S, et al. A vacuole-like compartment concentrates a disordered calcium phase in a key coccolithophorid alga. Nat Commun 7, 11228 (2016).
4. Jantschke A, Pinkas I, Schertel A, Addadi L, Weiner S. Biomineralization pathways in calcifying dinoflagellates: Uptake, storage in MgCaP-rich bodies and formation of the shell. Acta Biomaterialia 102, 427-439 (2020).
5. Jantschke A, et al. Anhydrous β -guanine crystals in a marine dinoflagellate: Structure and suggested function. J Struct Biol 207, 12-20 (2019).
6. Kadan Y, Aram L, Shimoni E, Levin-Zaidman S, Rosenwasser S, Gal A. In situ electron microscopy characterization of intracellular ion pools in mineral forming microalgae. J Struct Biol, 107465 (2020).
7. Paasche E. A review of the coccolithophorid *Emiliana huxleyi* (Prymnesiophyceae), with particular reference to growth, coccolith formation, and calcification-photosynthesis interactions. Phycologia 40, 503-529 (2002).
8. Yin X, et al. Formation and mosaicity of coccolith segment calcite of the marine algae *Emiliana huxleyi*. J Phycol 54, 85-104 (2018).
9. Didymus JM, Young JR, Mann S. Construction and Morphogenesis of the Chiral Ultrastructure of Coccoliths from the Marine Alga *Emiliana huxleyi*. Proceedings of the Royal Society of London Series B: Biological Sciences 258, 237-245 (1994).
10. Gerecht AC, Šupraha L, Langer G, Henderiks J. Phosphorus limitation and heat stress decrease calcification in *Emiliana huxleyi*. Biogeosciences Discuss 2017, 1-18 (2017).
11. Langer G, de Nooijer LJ, Oetjen K. On the role of the cytoskeleton in coccolith morphogenesis: The effect of cytoskeleton inhibitors. J Phycol 46, 1252-1256 (2010).
12. Westbroek P, et al. Mechanism of calcification in the marine alga *Emiliana huxleyi*. Phil Trans R Soc B 304, 435-444 (1984).
13. Hoffman DP, et al. Correlative three-dimensional super-resolution and block-face electron microscopy of whole vitreously frozen cells. Science 367, eaaz5357 (2020).
14. Engel BD, Schaffer M, Cuellar LK, Villa E, Pitzko JM, Baumeister W. Native architecture of the *Chlamydomonas* chloroplast revealed by in situ cryo-electron tomography. eLife 4, (2015).
15. Andersen RA, Saunders GW, Paskind MP, Sexton JP. Ultrastructure and 18s rRNA gene sequence for *Pelagomonas calceolata* gen. et sp. nov. and the description of a new algal class, the pelagophyceae classis nov. J Phycol 29, 701-715 (1993).

Reviewer #3 (Remarks to the Author):

This is the report to investigate the subcellular structure of different types of phytoplankton three-dimensionally using more than ten thousand of serial images obtained by FIB-SEM. Three dimensional subcellular structure of green algae was already reported using *Parachlorella kessleri* (Ota et al., 2016, *Biotechnol. Biofuels*, DOI: 10.1186/s13068-016-0424-2; Ota et al., 2016, *Sci. Rep.*, DOI: 10.1038/srep25731), which should be cited. In my knowledge, however, this is the first report to show unique subcellular architecture and compare the architecture between some different types of phytoplankton qualitatively and quantitatively. Thus, this study is novel and of interest for many researchers handling not only microalgae but also vascular plants. In addition, the authors used open-access software to analyze a large numbers of serial images, and the methodology may be useful for many researchers engaging in array tomography.

In Fig.3, the authors show and discuss the volume of the plastid, which occupied 30 to 40% of the cell volume. Recently, the volumetric analysis of mesophyll cells of higher plant of rice, wheat, and chick pea has been reported (see the below references). The percentages of the chloroplast volume in the mesophyll cell are 20 to 40% according to the references, though the percentage of the volume in rice is slightly higher than other plants. These results suggest that the plastid volume of photosynthetic cells could be similar, and thus it may be worth to discuss with the references.

(References)

Harwood et al., (2020) Cell and chloroplast anatomical features are poorly estimated from 2D cross-sections. *New Phytol.* 225(6): 2567-2578. DOI: 10.1111/nph.16219

Oi et al., (2017) Three-dimensional intracellular structure of a whole rice mesophyll cell observed with FIB-SEM. *Ann. Bot.* 120: 21-28. DOI: 10.1093/ab/mcx036

I found a few careless mistakes in the manuscript as described below.

- (1) P14/L27: "than ADP-glucose47)" → "than ADP-glucose47" (delete the parenthesis)
- (2) P17/L5: "(Fig. 7C, N°2 to N°5)" → "N°2 to N°5 (Fig. 7C)"

Referee's comments: black

Answer to comments: red

New text: green

In this manuscript, Uwizeye et al. use FIB-SEM imaging of resin-embedded cells to perform a survey of organelle morphology in six diverse species of phytoplankton. This work is a natural extension from this research group's previous FIB-SEM study of diatoms, which characterized physical interaction surfaces between mitochondria and the chloroplast (Flori et al., 2017; Nat Comm). 3D segmentation of the tomograms (3 cells examined per species) enabled quantification of the volumes occupied by various organelles (nucleus, mitochondria, chloroplast) and certain sub-organelle regions (pyrenoid, cristae, nucleolus). The cell biology and physiology of these phytoplankton species have not been well characterized before, so I think such descriptions are very important, both from an evolutionary perspective and also because these marine algae are major players in the global carbon cycle. As the Earth's climate rapidly warms, it is crucial to understand the cellular physiology of marine algae and how these algae respond to changing environmental conditions.

We would like to thank both Benjamin Engel and Wojciech Wietrzynsk for the very valuable and encouraging comments. In the new version of the manuscript, we have followed their advises to improve our work.

I am enthusiastic about this study's topic and FIB-SEM imaging approach. I enjoyed looking at the 3D cellular architecture revealed in the tomograms, which I feel provides a great starting point for future studies. However, I am disappointed that the current study does not go further to investigate the physiology of these cells. The analysis is limited to basic correlations of organelle volumes and surface areas between species, but does not begin to address the important question of how organelle architecture is remodeled in response to environmental change. If the authors were able to add some analysis addressing this topic, I would find this study very compelling.

This is a really good point. After receiving the reviews, we decided to follow the recommendation of Referees 1 and test the possible link between physiological changes in phytoplankton and changes in their subcellular architecture. We have been able to relate acclimation responses to structural modifications of the plastids and mitochondria volumes, areas and possible contacts points in *Phaeodactylum tricorutum* cells acclimated to two different illumination conditions (new Fig. 6) and in *Nannochloropsis gaditana* cells grown in two trophic conditions (new Fig. 7). Thanks to these findings, we can propose the existence of a link between acclimation responses and structural modifications in the energy making organelles.

Major Points:

1) This paper lacks a strong biological focus. The authors tease in their abstract that their approach "opens new perspectives to study acclimation responses to abiotic and biotic factors at a relevant biological scale", but they perform no experiments along these lines. I very much appreciate the long time it takes to perform the FIB-SEM analysis. However, I think the impact of this paper would be greatly increased if the authors explored at least one biological question in some detail. Some possibilities include changes in pyrenoid morphology in low vs. high CO₂ or changes in thylakoid-mitochondria contacts in different environmental conditions.

As mentioned above, we followed this advice, focusing on light acclimation in *Phaeodactylum* and acclimation to trophic lifestyles in *Nannochloropsis*, a new species that we added to our portfolio. We hope that the results meet the expectation of the Referees.

2) Part 1 of the results section (including Fig. 1) presents a detailed technical description of how the tomograms were segmented and the volumes / surface areas were measured. This part of the manuscript reads very much like a methods section, and thus should be moved to the methods. Fig. 1 should be a supplemental figure.

We also followed this advice, and moved this part of the text to the methods section. Fig. 1 is now supplementary Fig. 1. The text describing the imaging approach has been revised, also to address the comments of Referee 2. Two supplementary datasets have been made, containing the python scripts for metric analysis and distance evaluation. A detailed protocol on image analysis has been prepared and will be submitted to Bio-protocol if the manuscript is accepted, as indicated in the instructions for authors of this journal.

3) While I appreciate that this time-intensive technique limits the available statistics, the authors should be careful when making general conclusions from a small dataset (3 cells were imaged per species). For example, the authors state several times that the variable mitochondria architecture reflects the organelle's dynamic nature. Inferring dynamics requires either live-cell microscopy or at least far more static snapshots than were performed here, ideally with multiple physiological conditions. If it isn't feasible to increase the sample size, the authors should be cautious with their generalizations.

Finding that mitochondria are dynamic organelles is not an original contribution from us. Here, we simply observe a large variability in the shape of mitochondria, which we interpret based on previous studies on eukaryotic cells. We have reformulated the text to clarify this point, and modified supplementary figure 2 with sections through cellular 3D volumes (line 'a'), to allow a better identification of mitochondria in *Emiliana*, as requested by Referee 2.

Page 7: 'Mitochondria were characterized by more variable shapes not only between species but also within cells of the same species (e.g. Supplementary Fig. 2 in the case of *Emiliana*). This diversity probably reflects the dynamic nature of these organelles, which frequently change their shape, undergo dislocations, fusion and fission in eukaryotes (Bereiter-Hahn and Vöth, 1994)'

4) Besides *Phaeodactylum*, all the other algae species showed <2% of the plastid in contact with mitochondria. The authors conclude this suggests "that the whole process could be dynamic throughout the life cycle and environmental conditions, possibly being mediated by molecular actors". In the absence of experimental evidence looking at these conditions, this general conclusion feels far too speculative. How do the authors know that this small contacting surface area represents a specific interaction mediated by molecular actors? To this reviewer, a very plausible explanation could be that these are random contacts that do not transfer energy between the organelles. If anything, the major contrast to diatoms seems to suggest this could be the case. But in the absence of additional data, this too is speculation.

Following this comment, we have carefully re-examined surface areas of contact between plastids and mitochondria using two distance criteria values: ≤ 30 nm, defining the 'average' contact distance, and ≤ 90 nm, defining the 'upper limit' for contact. Results are presented in the new Fig. 4. As expected, we found that the size of the contacting surface areas increased when employing the higher distance

criterion for the computation. At this distance, surface areas of contact become also evident in species where they were not found at 30 nm (*Pelagomonas* and *Nannochloropsis*). However, the localisation of the contact areas does not change using the 30 nm or the 90 nm distance criteria. Moreover, the size of the surface areas of contact changes under growth conditions that modify the functional relationship between respiration and photosynthesis. To us, these findings suggest that surface areas of contact are not randomly localised, possibly representing contact points between the organelles. Unfortunately, the FIB-SEM approach method has not enough resolution to prove that these contact points mediate any type of exchange. We have therefore reformulated the text to tune down our claim and clarify this point:

Page 9: 'Following this definition, we calculated surface areas of contact between plastids and mitochondria using two distance criteria values: ≤ 30 nm, defining the 'average' contact distance, and ≤ 90 nm, defining the 'upper limit' for contact. Surface areas of contact varied depending on the microalgal species. At ≤ 30 nm (Fig. 4a), surface areas of contact were found in all algae (up to 7.1 ± 0.1 % of the plastid surface being involved in contacts with mitochondria in *Phaeodactylum*), except for *Pelagomonas* and *Nannochloropsis* (0.1 ± 0.1 and 0.16 ± 0.3 %, respectively). The surface area of contact largely increased when measured using the ≤ 90 nm distance criterion (Fig 4b). At this distance, surface areas of contact become detectable in *Pelagomonas* and *Nannochloropsis* (5.2 ± 0.4 % and 3.3 ± 1.7 %, respectively). In *Phaeodactylum*, these areas reached 15.7 ± 1.4 % of the plastid surface, when calculated at ≤ 90 nm. Nonetheless, contact areas maintained the same location on the plastid using both distance criteria. To us, this finding suggests that plastid-mitochondria interactions may occur at specific locations, as already proposed in the case of other organelles-organelle interactions (Phillips and Voeltz, 2016; Prinz, 2014; Rowland and Voeltz, 2012).'

5) The authors devote a large portion of the text to the pyrenoid. I agree that this is a fascinating and important topic that can be illuminated by FIB-SEM imaging. The pyrenoid has evolved convergently in each of these different species, so it is of high interest to carefully detail the variable morphology of these compartments. Unfortunately, it was hard for me to appreciate the pyrenoid architecture in the small segmentations provided in Fig. 6A. Perhaps the authors could create an additional figure showing the pyrenoid of each species in more detail, comparing slices through the raw tomograms with corresponding slices through the segmentations.

This figure (now Fig. 5), has been substantially modified, using larger panels to better highlight the structural features of the pyrenoid (Fig. 5a).

We have also modified the text, to remove speculative parts (e.g. the one discussing possible reasons to maintain starch close to the pyrenoid) and focus more on the structural features of the different pyrenoids.

pages 10-11: 'Plastids were mostly occupied by thylakoid membranes and the stroma, and by the carbon-fixing pyrenoid (Fig. 5b), a Rubisco-rich matrix that was absent in *Pelagomonas* (Andersen et al., 1993), *Galdieria* (Merola et al., 1981) and *Nannochloropsis* (Mackinder et al., 2016).

In two taxa (*Phaeodactylum* and *Emiliania*), we observed thylakoids crossing the pyrenoid matrix (Fig. 5a). These pyrenoid membranes (also called pyrenoid tubules in *Chlamydomonas* (Engel et al., 2015)) display different topologies: we observed parallel stacks in the diatom and a more branched structure in *Emiliania*, reminiscent of that reported in *Chlamydomonas* (Engel et al., 2015; Meyer et al., 2016). *Micromonas* and *Symbiodinium* contained thylakoid-free pyrenoids that were almost completely surrounded by starch sheaths (Fig. 5a). Few stalks ensure the connection between pyrenoid and the

plastid, possibly to facilitate the diffusion of Rubisco substrates and products as previously proposed (Badger and Price, 1994; Engel et al., 2015; Moroney and Mason, 1991), see also the review (Meyer et al., 2017). Unlike *Micromonas*, the pyrenoid of the dinoflagellate *Symbiodinium* was not centred in the plastid, but instead protruded towards the cytosol, being surrounded by a shell of cytosolic rather than stromal starch (Dauvillee et al., 2009; Meyer et al., 2017; Van Thinh et al., 1986).'

Minor Points:

Page 4: "we generate accurate 3D reconstructions of whole cells in their close-to-native states".
-- Even with the improved sample preservation provided by freeze substitution, the heavy metal staining and resin-embedding of cells performed in this study does not reveal cells in their "close-to-native state". This term should be reserved for hydrated frozen cells (e.g., cryo-EM).

The Referees are correct: we have removed this sentence

Page 8: "photosynthetic membranes (thylakoids) were organised in layers of a few stacks, without the typical differentiation in grana and stromal lamellae observed in vascular plants and green algae"
-- Green algae do not have the canonical grana and stroma lamellae of plants. This should be reworded to say something like "without the subdivision into stacked and unstacked membrane domains observed in green algae".

This is also true. Thus, we have removed 'green algae' from the sentence

Page 9: In Fig. 3C, the values of the normalized and absolute volumes do not match. For example, the plastid is the largest absolute volume in all species, but the cytosol is shown to occupy the largest percentage of the volume. Perhaps the color code was mixed up between the graphs (green switched with grey, red switched with blue)?

In the previous version of the manuscript, we made a mistake in assigning colours to the bars in this figure, where others inconsistencies were also present. In this new version, we have substantially modified this figure, to:

- i. include *Nannochloropsis cells*, since this species is now studied in the manuscript;
- ii. introduce cells in a more rational way (from the smallest to the largest one);
- iii. insert more appropriate sections through cellular 3D volumes (column a), since some of them were not corresponding to the central section of the FIB-SEM image (column b);
- iv. present correct histograms (column c).

Page 10: "Altogether, the organelles (nuclei, plastids and mitochondria) filled a relatively constant fraction (40 to 55 %) of the total cell volume"

-- The Golgi and ER are also universally considered to be organelles. The authors say that the Golgi has "high variability in terms of volume occupancy" but then conclude generally that: "We interpret this conservation of organelle volumes and the variability of the other compartments as the signature of evolutionary constraints that preserve essential cellular functions". This should be

reworded. The Golgi is an organelle, not an “other compartment”. Specifically, the volume of the nucleus, mitochondria, and chloroplast appear to be relatively conserved between the studied species, but this is not true for all organelles.

This is also true. Unfortunately, due to the limits of the FIB-SEM resolution, it is hard to precisely quantify the Golgi and ER volumes in some tomograms, making it difficult to treat them as ‘separate’ compartments. Therefore, in the new version of the manuscript, we simply consider ‘main organelles’ (nuclei, plastids and mitochondria) and ‘other’ compartments.

Page 10: “...unveiled the existence of a tight correlation between plastids and mitochondria in terms of their volume. This conserved topological feature corroborates the previous molecular and physiological evidence in diatoms for energetic interactions between plastids and mitochondria.”
-- It is not clear to me how a constant volume ratio between chloroplasts and mitochondria in several species corroborates energy exchange between these organelles.

This is again true. This sentence was certainly biased by previous results in diatoms. The text has been modified to acknowledge the fact that a relationship exists, and that it could play some role in organelle function at the level of energy management for carbon assimilation.

Our new data in diatoms exposed to low light and high light conditions (new Fig. 6) as well as the characterisation of *Nannochloropsis* cells grown in different nutrient environments (new Fig. 7) also support this conclusion.

We have modified the text at several places to discuss this possibility:

Page 9: ‘Plastid-mitochondria relationships are of primary importance in diatoms (Bailleul et al., 2015; Kim et al., 2016)), where interactions between the two organelles are relevant for carbon assimilation. Based on the findings above, it is possible that this organelle-organelle relationship also exists in other microalgal species’.

Page 14: ‘Indeed, previous work showed that organelle interactions is an advantage for carbon assimilation in diatoms, either to facilitate energetic interactions between the two cell engines (Bailleul et al., 2015), or to mediate lipid exchange, as proposed in plants (Mueller-Schuessele and Michaud, 2018)’.

Page 16: ‘We interpret this finding assuming that plastid-mitochondria contacts could facilitate energy exchanges between the organelles in *Nannochloropsis*, to readjust the balance between the two cell organelles according to the environmental conditions. Alternatively, contacts could mediate lipid exchange between plastids and mitochondria (Mueller-Schuessele and Michaud, 2018) to facilitate the structural changes induced by the two trophic conditions’.

Page 13: “Different levels of DNA condensation are visible. DNA is present in the form of compact chromosomes in Symbiodinium, possibly leaving a fraction of the nucleoplasm without chromatin (grey)”

-- How do the authors know that variable staining of the nucleoplasm with osmium tetroxide and uranyl acetate reflects degrees of DNA condensation? The resolution is far too limited to directly see chromatin compaction (i.e., nucleosomes). The authors should clearly state that this is an assumption they are making based on an indirect measurement (affinity for heavy metal stain), and they should cite literature that supports this assumption. The statement that some of the Symbiodinium nucleoplasm may lack chromatin is too speculative based on the resolution of this FIB-SEM data.

We reckon that this conclusion is not fully supported by the data. Therefore, we have decided to remove this part of the figure and the corresponding text.

Page 14: “The plastid volumes were mainly occupied by thylakoid membranes and, when present, by the pyrenoid”

-- When was the pyrenoid not observed? Do *Galdieria* and *Pelagomonas* lack pyrenoids? Is this expected based on previous literature or their phylogeny? It would be appropriate to have some discussion of why pyrenoids were absent. Is it the species, growth condition, or a technical limitation preventing detection?

The pyrenoid is not visible in *Galdieria*, *Nannochloropsis* and *Pelagomonas* consistent with previous reports. References are cited in the revised version of the manuscript to acknowledge the lack of pyrenoid in these species.

Page 10: ‘Plastids were mostly occupied by thylakoid membranes and the stroma, and by the carbon-fixing pyrenoid (Fig. 5b), a Rubisco-rich matrix that was absent in *Pelagomonas* (Andersen et al., 1993), *Galdieria* (Merola et al., 1981) and *Nannochloropsis* (Mackinder et al., 2016).’

We agree that the lack of pyrenoid in some algal species is intriguing. However, a discussion on this topic would be only speculative, and possibly out of the scope of this manuscript.

Page 17: “we foresee in the future a wide utilisation of this technique to analyse subcellular and even sub-organellar structures, to help understanding physiological responses at the macromolecular level.”

-- This should be reworded. FIB-SEM does not provide information at the macromolecular level.

the sentence has been removed

-- The number of replicates and statistical methods used should be described in every figure legend.

done now for every figure

-- The movie with the 3D tour of the coccolithophore segmentation is fantastic! I know it is labor-intensive, but I would love to see similar movies of the other algae.

Unfortunately, this has not been possible to make similar movies for the other algae. Nonetheless, we have decided to keep this video (now supplementary video 10) despite the fact that we removed the paragraph about coccolithogenesis (see answer to Referee 2 below), since the Referees consider this video important.

Formatting and Edits:

Page 2: “its diversity” -> their diversity

done

Page 3: “A high-throughput confocal” -> High-throughput confocal

done

Page 3: “However, optical microscopy studies have insufficient resolution to reveal microstructural features, emphasising the need to develop complementary imaging approaches to study the cellular and subcellular bases of phytoplankton physiology and cell biology.” -> However, optical microscopy studies have insufficient resolution to reveal cellular ultrastructure, emphasizing the need to develop complementary imaging approaches to study the structural basis of phytoplankton physiology.

This sentence no longer exists in the new version of the manuscript

Page 3: “3D electron microscopy (EM) imaging” -> 3D electron microscopy (EM)

This sentence no longer exists in the new version of the manuscript

Page 3: “to access the cellular and subcellular features of phytoplankton at the nanometric scale” -> to access the subcellular features of phytoplankton at the nanometer scale

This sentence no longer exists in the new version of the manuscript

Page 3: “providing contextual 3D images of whole cells at a specific time.” -> imaging 3D volumes of whole cells.

This sentence no longer exists in the new version of the manuscript

Page 5: “with a workable (~1000-1200) number of frames.” -> with ~1000-1200 frames.

This sentence no longer exists in the new version of the manuscript

Page 6 (Fig. 1 legend): “surface is obtained” -> surface area is obtained

done

Page 7: “e.g. coccolithophores in Emiliana” -> e.g. coccoliths in Emiliana (coccolithophores are the organisms, not the plate structures)

This sentence no longer exists in the new version of the manuscript

Page 8 (Fig. 2 legend): “External feature” -> External features

done

Page 9 (Fig. 3 legend): “cells as revealed by FIB-SEM” -> cells revealed by FIB-SEM

done

Page 9 (Fig. 3 legend): “Sections of cellular 3D volumes, based on FIB-SEM imaging” -> Sections through cellular 3D volumes, segmented from FIB-SEM images (or you could say tomograms)

done

Page 9 (Fig. 3 legend): “Images highlight” -> Segmentations highlight

done

Page 11 (Fig. 4 legend): “exegons” -> “Hexagons

done

Page 11 (Fig. 4 legend): “prevents a correct analysis” -> prevents a meaningful correlation

done

Page 12: “approximately 5% of the total surface” -> approximately 7% of the total surface (according to Fig. 5)

done

Page 13: “explore the sub-organelles features” -> explore the sub-organellar features

This sentence no longer exists in the new version of the manuscript

Page 15: “maintaining a proper balance between the subcompartments producing light-dependent energy (the photosynthetic membranes) and the light-independent one (the CO₂ fixing compartment).” -> maintaining a proper balance between the subcompartments performing light harvesting (the photosynthetic membranes) and CO₂ fixation (the pyrenoid).

done

Page 16: “become more structured insofar as the vesicles detached” -> “become more structured as the vesicles detach”

This sentence no longer exists in the new version of the manuscript

Page 17: “Since this technique can be applied to all type of cells” -> Since FIB-SEM can be applied to all types of cells

This sentence no longer exists in the new version of the manuscript

Reviewed by Benjamin Engel and Wojciech Wietrzynski

Reviewer #2 (Remarks to the Author):

The manuscript "In-cell quantitative structural imaging of phytoplankton using 3D electron microscopy" by Uwizeye et al. describes the cellular and subcellular architecture of six eukaryotic microalgae based on 3D-imaging data that was obtained by focused Ion Beam Scanning EM (FIB-SEM). From the imaging data, 3D models of cells and subcellular structures were generated and compared using three-dimensional (3D) morphometric analyses. The latter led to the hypothesis that the volumetric ratio of plastids and mitochondria may be constant in the different algae lineages. Although we lack 3D ultrastructural data for most of the here investigated algae and I find it of great importance to study ultrastructure of eukaryotic algae to better understand their biology, I have substantial concerns with respect to the developed workflow and the presentation and interpretation of the data

Thanks to Referee 2 for the detailed and compelling review.

We agree with her/him that our interpretation of the process of coccolithogenesis was superficial. Moreover, this part was not the 'core' of our manuscript. Since in this new version of the manuscript we focus more on the relationship between physiology and subcellular architecture (with an emphasis on the energy producing organelles plastids and mitochondria), we believe that the part on biomineralization is no longer needed. Therefore, we have entirely removed this figure (old Fig. 7) and the corresponding text from the manuscript. We only maintained the supplementary video (now Supplementary video 10), to illustrate the impact of making a 3D tour into a segmented cell (see e.g. the comment of Referee 1).

We reckon that the findings on coccolith architectures may suggest that cells were in a poor physiological status, as pointed out by Referee 2. However, we already had evidences (based on chlorophyll fluorescence measurements) suggesting that this was not the case when the manuscript was first submitted, thus addressing the second main criticism Referee 2. These data are now presented in the supplementary Table 1, where a comparison is made with previously published data in the literature.

In the following, we present a more detailed response to her/his criticisms

In recent years, volume microscopy has become an important method in cell biology. I am aware of whole-cell 3D FIB-SEM data of a few celled photosynthetic organisms [1-5] (see below for references). To my surprise, neither the *Chlamydomonas reinhardtii* [1,2], *Emiliana huxleyi* [3], nor the dinoflagellate *Calciodinellum operosum* [4, 5] work is referenced at the appropriate position in the introduction where the authors state "This technique (FIB-SEM) has already been applied successfully to provide 3D models of eukaryotic cell" (page 3).

This part of the introduction has been substantially revised, to provide a more complete state of the art in the field. References suggested by Referee 2 and 3 are now quoted in the text along with other references on 3D imaging in plants and algae. We decided to maintain a focus on photosynthetic organisms, since they are the mainly focus of this work.

Pages 3-4: 'Thanks to the recent development of 3D Electron Microscopy (EM) methods (Baumeister 2002; Phan et al., 2016; Titze and Genoud 2016), 3D reconstructions have been obtained to analyse plant cell division (Mineyuki 2014), chloroplast biogenesis (Liang et al., 2018), with emphasis on thylakoids organization (Shimoni et al., 2005; Daum et al., 2010 Austin and Staehelin, 2011; Kouřil et al., 2011; Kowalewska et al., 2016) and algal cell structures (Moisan et al., 1999; 2006; Wayama et al., 2013; Ota et al., 2016a; Ota et al., 2016b). Serial Block Face Scanning Electron Microscopy (SFB-SEM) has been used to analyse plant subcellular architectures (Kittelmann et al., 2016; Pain et al., 2019; Harwood et al., 2020). Ion-beam milling was used to prepare thin lamella for imaging by cryo-EM (Rigort et al., 2012), revealing the native architecture of the *Chlamydomonas* chloroplast (Engel et al., 2015; Shaffer et al., 2015; Wietrzynski et al., 2020). Focused Ion Beam Scanning Electron Microscopy (FIB-SEM) has been used to reveal the 3D structure of photosynthetic cells with enough resolution (4-10 nm) to investigate their subcellular architecture. This technique has been applied to chemically fixed samples in rice (Oi et al., 2017; Yamane et al., 2018), *Chlamydomonas* (Xu et al., 2017; García-Cerdán et al., 2020), and in the diatom *Phaeodactylum tricornutum* (Flori et al., 2016; 2017), and to cryo-fixed and freeze substituted cells (Decelle et al., 2019). Cryo-FIB-SEM of high-pressure frozen marine algae such as coccolithophores (Sviben et al., 2016) and dinoflagellates (Jantschke et al., 2019; 2020) has also been used to study biomineralization pathways'.

In contrast to the author's work, previous 3D imaging of *Emiliana* was based on cryo FIB-SEM. Cryo FIB-SEM does not involve artifact introducing sample preparation steps as does the FIB-SEM workflow used by the authors. With all due respect to the large quantity of data and the challenging work produced, the authors should know that their sample preparation protocol does not preserve soluble calcium phases [6] and thus data sets from cells that have been prepared by the author's method lack important information. This limitation of the method and an apparent lack of knowledge about the ultrastructure of *E. huxleyi* and the process of coccolith formation led to severe misinterpretation of the cellular structures that can be seen in the *Emiliana* data set. Firstly, *Emiliana* forms coccoliths inside the cells in a membrane-bound compartment called the coccolith vesicle. Usually, one coccolith is formed at a time and always one coccolith in one coccolith vesicle [7]. Instead, the authors write: "... formation of a proccoloth (do you mean protococcolith?) within Golgi-associated vesicles ..." (page 16). Unequivocal identification of the coccolith vesicle is only possible when it contains a nascent coccolith. Without containing a nascent coccolith the coccolith vesicle is hardly distinguishable from other membrane-bound structures. Neither in the SEM images shown in figure 7 nor the video in the supplementary material a cell with a nascent coccolith within a coccolith vesicle is visible. Therefore, I have no idea where the authors have seen the coccolith vesicle, protococcolith, and organic base plate scale for which they generated the 3D models shown in figure 7b. All I can see is that these models lack the morphology that one would expect for these structures [3, 8, 9]. The authors should provide the SEM images/image stacks, and not only 3D models. Mistakes in the identification of subcellular structures can then be corrected. The structure of which

the authors claim to be the Golgi apparatus seems to me to be the reticular body [8], a complex membrane system positioned on top of the coccolith vesicle in *Emiliana*. The reticular body is in close contact with the coccolith vesicle and not the Golgi apparatus, as the authors write on page 16. The reticular body due to its large size a conspicuous organelle. The Golgi apparatus in *Emiliana* is usually smaller than the reticular body and easy to overlook. Secondly, once calcification in the coccolith vesicle is complete, the mature coccolith is secreted onto the cell surface. The coccosphere should therefore mainly contain mature coccoliths. Immature coccoliths are usually secreted to the cell surface only under stress conditions [10, 11]. The fact that the authors observe multiple

immature coccoliths on the cell surface suggests that the cells experienced severe stress for several hours before they were snap frozen. Under ideal conditions it takes about 1 hour to form a coccolith [7]. The impact of stress on the volumes of organelles has not been investigated to my knowledge. I think this aspect should be clarified before comparing stressed *Emiliania* cells with cells of species that may be in a different physiological state. Thirdly, the nucleation of calcite crystals inside the coccolith vesicle takes place on the surface of a disc-shaped organic structure, which is called base plate or “organic base plate scale”. The base plate scale becomes a component of the coccolith. It is usually visible inside the coccolith vesicle before the onset of calcite nucleation and at the base of nascent and mature coccoliths [12]. Therefore, the statement of the authors “... coccoliths are released outside the cells making the Organic Base Plate Scale (OBOS) easy detectable in the cell.” (figure 7 legend) makes no sense to me. In none of the provided images, the coccolith base plate scale is visible to me. By the way, reference 49 is wrongly claimed to be a review of coccolithogenesis (page 16).

We agree with the conclusion made by Referee 2 that it is not possible to draw precise conclusions on the coccolithogenesis from our data. However, the only purpose of this figure was to illustrate a possible application of FIB-SEM to study biomineralization, in parallel to the analysis of organelles metrics. Following the advises of Referee 1, we have now substantially changed the focus of the manuscript. We consider therefore that this controversial part is no longer relevant and we have removed old Fig. 7 and the entire paragraph describing these data in the new version of this manuscript.

Another concern I have is related to the quantification of the contact areas between organelles. The authors define membranes to be in contact when their distance is less than 30 nm (page 12). Even though there is no accepted definition for contact sites, a 30 nm gap is a huge distance. Ribosomes have a diameter of ~20 nm and therefore would fit into a 30 nm gap. Ribosomes between the membranes of two organelles would speak against contact.

The criteria employed here to define contact areas are taken from a ‘consensus’ article published by community working in this field (Scorrano et al., Nat commun 2019). We merely followed their recommendation. In the new version of the manuscript, we used two distance criteria to evaluate surface areas of contact: ≤ 30 nm, defining the ‘average’ contact distance, and ≤ 90 nm, defining the ‘upper limit’ for contact. We found that *i.* surface areas of contact increased in size when comparing ≤ 90 and ≤ 30 nm, *ii.* even at the highest distance, there are no random surface areas of contact appearing. Moreover, *iii.* surface areas of contact ‘respond’ to changes in the growth conditions increasing the light intensity in *Phaeodactylum*, and changing the trophic conditions in *Nannochloropsis*. To us, these findings suggest that we are observing a real biological phenomenon and not an artefact. As already acknowledged above for Referee 1, the biological meaning of these contact points (energy exchanges, metabolism, etc) cannot be evinced from our measurement. Therefore, we only propose a hypothetical explanation for their function, based on earlier suggestions in plants (Mueller-Schuessele and Michaud, 2018) and algae (Bailleul et al., 2015).

Page 14: ‘previous work showed that organelle interactions is an advantage for carbon assimilation in diatoms, either to facilitate energetic interactions between the two cell engines (Bailleul et al., 2015) 5, or to mediate lipid exchange, as proposed in plants (Mueller-Schuessele and Michaud, 2018)’.

Page 16: ‘We interpret this finding assuming that plastid-mitochondria contacts could facilitate energy exchanges between the organelles in *Nannochloropsis*, to readjust the balance between the two cell organelles according to the environmental conditions. Alternatively, contacts could mediate lipid

exchange between plastids and mitochondria (Mueller-Schuessele and Michaud, 2018) to facilitate the structural changes induced by the two trophic conditions’.

Unfortunately, the quality and resolution of the images/videos provided is so poor that ribosomes cannot be seen. That FIB-SEM can visualize ribosomes has been shown previously (figure 5B reference [2]). In comparison to the presentation of previous FIB-SEM work in other high-ranking journals [2, 13, 14], I find the presentation of the data in this manuscript poor. Given that FIB-SEM is the only method used in this study, the data should be presented in a more detailed and precise way, making it possible for the reader to check the author’s interpretation of the data. For example, the supplementary figure 1 claims to show 3D reconstructions of mitochondria of *Emiliana*, but the corresponding images to dispel doubts that this membrane system is peripheral endoplasmic reticulum are missing.

Videos illustrating the original FIB-SEM frames are available for all the algal strains/conditions investigated in this work (supplementary videos 1-9). We have decreased the frame rate of all videos to facilitate the analysis of the data. Stacks will be made available upon request if the manuscript is accepted.

Sections through cellular 3D volumes have been modified in Fig. 2 (column a, see also our answer to Referee 1 above), since some of them were not corresponding to the central section of the FIB-SEM image (column b).

Sections through cellular 3D relative to Fig. 5 are presented in Supplementary Fig. 4 (these images were already available in the old version of this manuscript). We also added sections through cellular 3D relative to Supplementary Fig. 2 as new panels (line ‘a’), again to allow a direct comparison between EM data and 3D reconstructions (line ‘b’).

Overall, these changes should make possible for the reader to check our interpretation of the data

The presented 3D model of *Pelagomonas calceolate* (fig 3, fig 4) is not in line with the original description of the species. The cells are described to be flagellated, elliptically shaped and to have a single chloroplast [15]. Has it been checked by sequencing of genetic marker genes that the received strain is truly *Pelagomonas calceolate*? It would not be the first time that a culture collection has send a wrong species.

We have not tested the real identity of our strains, but merely trusted the information provided with the culture. Therefore, we now present them based on the name that was provided to us by the culture collections (mostly the culture collection of Roscoff, France). This will allow readers to easily identify them.

I strongly recommend adding physiological data of the cells that were subjected to imaging in future manuscript submissions to demonstrate that the cells of the different species are in a similar physiological state. In find this important for the interpretation of the volume occupancy of organelles. For example, it is well established that N-starvation induces the formation of lipids droplets in algae. Lipid droplets are therefore not a specific subcellular feature of *Phaeodactylum tricornutum* (page 7).

We now provide measurements of the photosynthetic parameter F_v/F_m (assessing the photosynthetic efficiency of Photosystem II) for all the strains explored in this work. This parameter is widely used to

test the physiological conditions of the algae (e.g. it is a very good proxy of nitrogen starvation in diatoms, Abida et al., Plant physiol, 2015, 167: 118-136).

As show in supplementary table 1, the values measured in our samples prior to cryofixation are in good agreement with previous reports in the same species (when available) or closely related species.

We agree that the presence of lipid droplets in *Phaeodactylum* is not a specific feature of this alga. Indeed, they are more abundant in another stramenopile/heterokonta member -*Nannochloropsis*- even in nutrient replete conditions, in agreement with data from some of us (Simionato et al., Eukaryot Cell 2013 12:665-676) and other groups.

Finally, the workflow developed by the authors is by far not sufficiently detailed to be reproduced. Details of the filter parameter should be given and the Python script for filtering should be included to the supplementary material.

The description of the workflow (which has now been moved to the methods section following the advises of Referees 1) has been revised. As already mentioned in our answer to Referee 1, a complete workflow for image analysis will be submitted to Bio-protocol if the manuscript is accepted, as indicated in the instructions for authors of this journal.

The python scripts have been added to the new version of the manuscript as supplementary data 2 (for metric computations) and supplementary data 3, (distance computation between two meshes).

In summary, this is an interesting study, but as explained above I cannot verify the conclusions of the authors because the data is not presented in a precise and detailed way. The reported data on *Emiliana* contradict in many aspects the current knowledge on coccolith formation in this species. This misinterpretation of data makes this work not very convincing. The legend of figure 7 and last paragraph before the conclusion section should be completely revised.

As mentioned above, we now provide more information about the methods. New sections through cellular 3D volumes are shown. The frame rate of the supplementary videos has been reduced to facilitate the analysis of the data.

References:

1. García-Cerdán JG, et al. Chloroplast Sec14-like 1 (CPSFL1) is essential for normal chloroplast development and affects carotenoid accumulation in *Chlamydomonas*. PNAS 117, 12452-12463 (2020).
2. Xu CS, et al. Enhanced FIB-SEM systems for large-volume 3D imaging. eLife 6, e25916 (2017).
3. Sviben S, et al. A vacuole-like compartment concentrates a disordered calcium phase in a key coccolithophorid alga. Nat Commun 7, 11228 (2016).
4. Jantschke A, Pinkas I, Schertel A, Addadi L, Weiner S. Biomineralization pathways in calcifying dinoflagellates: Uptake, storage in MgCaP-rich bodies and formation of the shell. Acta Biomaterialia 102, 427-439 (2020).
5. Jantschke A, et al. Anhydrous β -guanine crystals in a marine dinoflagellate: Structure and suggested function. J Struct Biol 207, 12-20 (2019).
6. Kadan Y, Aram L, Shimoni E, Levin-Zaidman S, Rosenwasser S, Gal A. In situ electron microscopy characterization of intracellular ion pools in mineral forming microalgae. J Struct Biol, 107465 (2020).
7. Paasche E. A review of the coccolithophorid *Emiliana huxleyi* (Prymnesiophyceae), with particular reference to growth, coccolith formation, and calcification-photosynthesis interactions. Phycologia 40, 503-529 (2002).

8. Yin X, et al. Formation and mosaicity of coccolith segment calcite of the marine algae *Emiliana huxleyi*. *J Phycol* 54, 85-104 (2018).
9. Didymus JM, Young JR, Mann S. Construction and Morphogenesis of the Chiral Ultrastructure of Coccoliths from the Marine Alga *Emiliana huxleyi*. *Proceedings of the Royal Society of London Series B: Biological Sciences* 258, 237-245 (1994).
10. Gerecht AC, Šupraha L, Langer G, Henderiks J. Phosphorus limitation and heat stress decrease calcification in *Emiliana huxleyi*. *Biogeosciences Discuss* 2017, 1-18 (2017).
11. Langer G, de Nooijer LJ, Oetjen K. On the role of the cytoskeleton in coccolith morphogenesis: The effect of cytoskeleton inhibitors. *J Phycol* 46, 1252-1256 (2010).
12. Westbroek P, et al. Mechanism of calcification in the marine alga *Emiliana huxleyi*. *Phil Trans R Soc B* 304, 435-444 (1984).
13. Hoffman DP, et al. Correlative three-dimensional super-resolution and block-face electron microscopy of whole vitreously frozen cells. *Science* 367, eaaz5357 (2020).
14. Engel BD, Schaffer M, Cuellar LK, Villa E, Plitzko JM, Baumeister W. Native architecture of the *Chlamydomonas* chloroplast revealed by in situ cryo-electron tomography. *eLife* 4, (2015).
15. Andersen RA, Saunders GW, Paskind MP, Sexton JP. Ultrastructure and 18s rRNA gene sequence for *Pelagomonas calceolata* gen. et sp. nov. and the description of a new algal class, the pelagophyceae classis nov. *J Phycol* 29, 701-715 (1993).

many thanks for suggesting these references, which are now quoted in the manuscript.

Reviewer #3 (Remarks to the Author):

This is the report to investigate the subcellular structure of different types of phytoplankton three-dimensionally using more than ten thousand of serial images obtained by FIB-SEM. Three dimensional subcellular structure of green algae was already reported using *Parachlorella kessleri* (Ota et al., 2016, *Biotechnol. Biofuels*, DOI: 10.1186/s13068-016-0424-2; Ota et al., 2016, *Sci. Rep.*, DOI: 10.1038/srep25731), which should be cited. In my knowledge, however, this is the first report to show unique subcellular architecture and compare the architecture between some different types of phytoplankton qualitatively and quantitatively. Thus, this study is novel and of interest for many researchers handling not only microalgae but also vascular plants. In addition, the authors used open-access software to analyze a large numbers of serial images, and the methodology may be useful for many researchers engaging in array tomography.

Many thanks to Referee 3 for her/his positive analysis.

In Fig.3, the authors show and discuss the volume of the plastid, which occupied 30 to 40% of the cell volume. Recently, the volumetric analysis of mesophyll cells of higher plant of rice, wheat, and chick pea has been reported (see the below references). The percentages of the chloroplast volume in the mesophyll cell are 20 to 40% according to the references, though the percentage of the volume in rice is slightly higher than other plants. These results suggest that the plastid volume of photosynthetic cells could be similar, and thus it may be worth to discuss with the references.

Many thanks for this interesting suggestion. Following the advises of Referee 3, we have now mentioned the similarity between plants and phytoplankton cells in the results/discussion section (page 7).

Page 7: 'Quantitative analysis indicates that plastids always occupied the largest fraction (25-40 %) of the cell (Fig. 2c, Supplementary Dataset 1) in line with recent estimates in vascular plants (Oi et al., 2017; Harwood et al., 2020), followed by the nucleus (5-15 %) and the mitochondria (2.5 to 5%)'.

(References)

Harwood et al., (2020) Cell and chloroplast anatomical features are poorly estimated from 2D cross-sections. *New Phytol.* 225(6): 2567-2578. DOI: 10.1111/nph.16219

Oi et al., (2017) Three-dimensional intracellular structure of a whole rice mesophyll cell observed with FIB-SEM. *Ann. Bot.* 120: 21-28. DOI: 10.1093/ab/mcx036

many thanks for suggesting these references, which are now quoted in the manuscript.

I found a few careless mistakes in the manuscript as described below.

(1) P14/L27: "than ADP-glucose47)" → "than ADP-glucose47" (delete the parenthesis)

done

(2) P17/L5: "(Fig. 7C, N^o2 to N^o5)" → "N^o2 to N^o5 (Fig. 7C)"

This figure no longer exists.

Short summary of the changes made to the manuscript:

Changes in figures, videos, tables and datasets.

Figure 1: has been transferred to the supplementary data as suggested by referee 1. The figure has been modified to improve its clarity: panel 'e' has been changed to better illustrate how meshes are used to calculate surfaces and volumes.

Figure 2 (now Figure 1): unchanged

Figure 3 (now Figure 2): i. we inserted new data concerning *Nannochloropsis*, since this species is now studied in the manuscript. ii. We present cells in a more rational way (from the smallest to the largest one). iii. Column 'a' now contains new sections through cellular 3D volumes, since some of them were not corresponding to the central section of the FIB-SEM image (column 'b'). iv. Column 'c' now presents correct histograms since, as pointed out by referee 1, errors were present in the previous version.

Figure 4 (now Figure 3): we added data concerning *Nannochloropsis*.

Figure 5 (now Figure 4): surface areas of contact are now presented using a new colour (magenta) to better highlight them. Following advised by referees 1 and 2, we have now implemented the procedure to evaluate these areas, using two distance criteria (only one was used in the old version). Therefore, the figure contains two lines: 'a' distance criterion ≤ 30 nm, defining the 'average' contact distance between interacting organelles. Line 'b': distance criterion ≤ 90 nm, defining the 'upper limit' for contacts between interacting organelles, according to *Scorrano et al.* (ref 55 in the new manuscript).

Figure 6 (now Figure 5): we removed line 'a' presenting data on the chromatin status in the nucleus (due to the doubts casted by referee 1). We replaced it with a more detailed presentation of the pyrenoid (new line 'a'), using enlarged panels to allow readers to better appreciate their structural features (again following referee 1's advises).

Figure 7: removed since the information provided by this figure was highly questionable, as pointed out by referee 2.

Two new figures have been added to highlight the intimate links between cellular structures, energy balance and physiological responses, as advised by referee 1.

Figure 6: illustrates the link between physiological changes and subcellular modifications in *Phaeodactylum* cells grown in low light and high light. Photophysiology data (panel 'c'), 3D images (panel 'a') and quantitative analysis of morphological changes (panel 'b' and 'd') are presented to draw a scenario of light acclimation in this diatom.

Figure 7: illustrates the link between physiological changes and subcellular modifications in *Nannochloropsis* cells grown in autotrophic (light as the only energy source) and mixotrophic (light plus exogenous carbon) conditions. Again, photophysiology data (panels 'b' and 'c') are shown along with EM pictures (panel 'a'), 3D images (panel 'd') and quantitative analysis of morphological changes (panel 'e') to interpret acclimation of this alga to different trophic lifestyles in terms of subcellular structural changes.

Supplementary Figure 1: this is old Figure 1 (see above)

Supplementary Figure 2: we have introduced sections through cellular 3D volumes (line 'a'), to allow a better identification of mitochondria in *Emiliania*, as requested by Referee 2

Supplementary Figure 3: we introduced data on *Nannochloropsis* cells.

Supplementary Figure 4: unchanged

Supplementary table 1: we included measurements of the physiological status of the algae (through the photosynthetic parameter Fv/Fm), and a comparison with data from the literature. This parameter is widely used to test the physiological conditions of the algae, and our measurements indicate that samples were in an active, unstressed state before cryofixation.

3 new supplementary videos have been added to show features of *Nannochloropsis* cells in both autotrophic (supplementary video 3) and mixotrophic (supplementary video 9) conditions and features of *Phaeodactylum* cells grown in high light (supplementary video 8). The frame rate of all videos has been reduced to allow a better appreciation of single frames, as requested by referee 2.

Two supplementary datasets have been added to list the python scripts used to calculate subcellular surfaces and volumes (supplementary dataset 2) and contact surface areas between plastids and mitochondria (supplementary dataset 3), as requested by referee 2.

Changes in the text and the supplementary information.

The text has also been largely reviewed, to accommodate the changes in figures and tables described above. In particular:

the title and abstract have been revised to better focus on the new message of the manuscript: highlight the link between physiological changes in phytoplankton and subcellular modifications in organelles devoted to energy management.

The introduction has been modified following advises by referee 2, with a more detailed of the state of the art concerning 3D imaging in phototrophs (including plants as suggested by referee 3) by FIB-SEM, SFB-SEM and cryo ET.

The results and discussion section has been extensively modified to illustrate and comment the new data. A detailed list of the change introduced in the text is provided in the rebuttal letter. The paragraph concerning coccolithogenesis in *Emiliana* has been entirely removed, since we agree with referee 2 that this part was superficial. However, this part was not representing the 'core message' of our work. By removing it, we are now able to propose a manuscript that is more focused and provides more compelling evidences for the link between phytoplankton acclimation and subcellular modification in energy-making organelles.

The conclusion has been entirely revised, again to provide a more concise and convincing message.

References have been implemented in terms of quantity (to better cover the topics) and quality (to better highlight specific concepts). Overall, we introduced 27 new references in the main text and 9 in the supplementary information.

REVIEWER COMMENTS

Reviewer #1 (Remarks to the Author):

Reviewed by Benjamin Engel and Wojciech Wietrzynski

In their revised manuscript, Uwizeye et al. have added new investigations into the effects of changing environmental conditions on phytoplankton organelle architecture. Specifically, they examine acclimation of *Phaeodactylum* to two light intensities and the growth of *Nannochloropsis* in phototrophic vs. mixotrophic conditions. The *Nannochloropsis* remodeling is particularly striking, and is sure to spur new lines of inquiry. These new experiments strive to address my major issue with the original manuscript, and I believe the study is now a stronger candidate for publication. This paper supports the important goal of understanding the cellular and metabolic responses of globally important algae to the Earth's rapidly changing climate.

However, I have a remaining major issue with the analysis. In Fig. 4, the authors chose to define contact sites between chloroplasts and mitochondria using two distance metrics: <30 nm and <90 nm. I understand that the resolution of FIB-SEM imaging is limited, but I don't see how a separation of 90 nm between the organelles can be considered a contact site. In cryo-electron tomography, our group has observed specific contact sites between thylakoids and the plasma membrane with a ~ 3 nm intermembrane space (<https://pubmed.ncbi.nlm.nih.gov/30962530/>). Using the same technique, ER-plasma membrane, ER-mitochondria, and nucleus-vacuole contact sites were measured with intermembrane distances of ~ 20 nm, ~ 10 nm, and ~ 15 nm, respectively (<https://pubmed.ncbi.nlm.nih.gov/31743662/>). Even with the limited resolution of classical electron microscopy, ER-plasma membrane contact sites were confirmed to have ~ 20 nm spacing (<https://pubmed.ncbi.nlm.nih.gov/31743663/>), so I do not think this is a limitation of the technique. Thus, <30 nm seems to be a reasonable metric for chloroplast-mitochondria contact sites, but I have a hard time imagining protein structures that would span 90 nm to maintain a specific contact site between organelles. I would gladly reconsider my position on this if the authors can provide examples from the literature of bonafide contact sites with 90 nm intermembrane distances.

With the <30 nm threshold, the only species of algae that appears to have substantial contact ER-mitochondria contact sites is *Phaeodactylum*, which was already described in the authors' previous publication. There is no way to tell from the current data whether the $<2\%$ contact surface in the other organisms is a specific interface or just rare places where the organelles happened to be randomly close together. This is ok. The data should still be presented, but the authors should interpret it for what it is. Other than in diatoms, it appears that they do not see evidence for significant ER-mitochondria contact sites (at least under these growth conditions).

Typos:

Pg. 3: "cellular (subcellular) architecture" -> cellular and organellar architecture

Pg. 3: "2 dimensional (2D)" -> two dimensional (2D)

Pg. 3: "3-dimensional (3D)" -> three dimensional (3D)

Pg. 4: "phytoplankton representative" -> phytoplankton representatives

Pg. 14: "organelle interactions is an advantage" -> organelle interactions are an advantage

Reviewer #2 (Remarks to the Author):

I am fully satisfied with all changes made by the authors and recommend this manuscript for publication in Nature Communications.

Minor comment:

The authors may want to modify the sentence "Prior to FIB-SEM imaging, live cells were tested for photosynthetic capacity to verify their physiological status" (page 4) as the current wording is not precise enough regarding if multiple individual cells or culture aliquots (a mix of cells in different physiological states) were analyzed and what "physiological status" the cells had to have in order to be further processed for imaging.

Referee's comments: black

Answer to comments: red

New text: green

REVIEWER COMMENTS

Reviewer #1 (Remarks to the Author):

Reviewed by Benjamin Engel and Wojciech Wietrzynski

In their revised manuscript, Uwizeye et al. have added new investigations into the effects of changing environmental conditions on phytoplankton organelle architecture. Specifically, they examine acclimation of *Phaeodactylum* to two light intensities and the growth of *Nannochloropsis* in phototrophic vs. mixotrophic conditions. The *Nannochloropsis* remodeling is particularly striking, and is sure to spur new lines of inquiry. These new experiments strive to address my major issue with the original manuscript, and I believe the study is now a stronger candidate for publication. This paper supports the important goal of understanding the cellular and metabolic responses of globally important algae to the Earth's rapidly changing climate.

Many thanks again to Benjamin Engel and Wojciech Wietrzynski for their very useful comments, which allowed us to further improve the manuscript.

However, I have a remaining major issue with the analysis. In Fig. 4, the authors chose to define contact sites between chloroplasts and mitochondria using two distance metrics: <30 nm and <90 nm. I understand that the resolution of FIB-SEM imaging is limited, but I don't see how a separation of 90 nm between the organelles can be considered a contact site. In cryo-electron tomography, our group has observed specific contact sites between thylakoids and the plasma membrane with a ~3 nm intermembrane space (<https://pubmed.ncbi.nlm.nih.gov/30962530/>). Using the same technique, ER-plasma membrane, ER-mitochondria, and nucleus-vacuole contact sites were measured with intermembrane distances of ~20 nm, ~10 nm, and ~15 nm, respectively (<https://pubmed.ncbi.nlm.nih.gov/31743662/>). Even with the limited resolution of classical electron microscopy, ER-plasma membrane contact sites were confirmed to have ~20 nm spacing (<https://pubmed.ncbi.nlm.nih.gov/31743663/>), so I do not think this is a limitation of the technique. Thus, <30 nm seems to be a reasonable metric for chloroplast-mitochondria contact sites, but I have a hard time imagining protein structures that would span 90 nm to maintain a specific contact site between organelles. I would gladly reconsider my position on this if the authors can provide examples from the literature of bonafide contact sites with 90 nm intermembrane distances.

With the <30 nm threshold, the only species of algae that appears to have substantial contact ER-mitochondria contact sites is *Phaeodactylum*, which was already described in the authors' previous publication. There is no way to tell from the current data whether the <2% contact surface in the other organisms is a specific interface or just rare places where the organelles happened to be randomly close together. This is ok. The data should still be presented, but the authors should interpret it for what it is. Other than in diatoms, it appears that they do not see evidence for significant ER-mitochondria contact sites (at least under these growth conditions).

This is a really good point. Following the referees' advises we have substantially revised this part of the manuscript. As suggested, we kept the data, but discussed independently results obtained with the two distance criteria.

We first consider data relative to the ≤ 30 nm distance as representative of possible organelle contact points. To justify this choice, we quoted the references suggested by the referees (new references 54-56). Using this criterion, only *Phaeodactylum* cells display organelle-organelle contacts.

Next, we discuss the results obtained using the ≤ 90 nm distance criterion, as a possible signature of organelle proximity. We propose that organelle proximity is a shared feature of the different cells studied in this work.

Pages 6-7: Plastid-mitochondria interactions may rely on physical interactions between the two organelles (Flori et al., 2017; Mueller-Schuessele and Michaud, 2018). We tested this possibility by quantifying possible contact points between plastids and mitochondria in the different species analysed above (Fig. 4). Recent work based on cryo-electron tomography of cyanobacterial cells has revealed specific contact sites between thylakoids and the plasma membrane with a ~ 3 nm intermembrane space (Rast et al. 2019). Using the same technique, ER-plasma membrane, ER-mitochondria, and nucleus-vacuole contact sites were measured in eukaryotic cells with intermembrane distances of ~ 20 nm, ~ 10 nm, and ~ 15 nm, respectively (Collado, et al. 2019; Hoffmann et al., 2019). Based on these results, we chose a distance value of ≤ 30 nm to calculate surface areas of contact between plastids and mitochondria. We could identify contacts in *Phaeodactylum* (7.1 ± 0.1 % of the plastid surface being involved in contacts with mitochondria, Fig. 4a), in agreement with previous suggestions (Bailleul et al., 2015). Conversely, contacts turned out to be almost negligible in all the other organisms, ranging from 0.1 ± 0.1 in *Pelagomonas* to 1.8 ± 0.9 % of the plastid surface in *Emiliana*.

Other distance criteria have been proposed to operationally track contact points between organelles in light/EM microscopy (Scorrano et al., 2018). Distances ≤ 90 nm may represent an 'upper limit' for contacts. Using this criterion, areas became larger in *Phaeodactylum* (15.7 ± 1.4 % of the plastid surface Fig. 4b), and evident in all the tested organisms. However, due to the quite large intermembrane distance, areas calculated with this criterion likely represent a proximity between plastids and mitochondria, rather than genuine contact sites between the two organelles mediated by protein machineries, as observed in the case of other organelles-organelle interactions (Phillips et al., 2016; Prinz et al., 2014; Rowland et al., 2012).

The concept of organelle proximity has been employed throughout the entire text, and in Figures 4 and 6 to discuss changes in the relative distance between plastids and mitochondria during acclimation to light in *Phaeodactylum* and to different trophic lifestyles in *Nannochloropsis*.

Page 10: Plastid-mitochondria proximity increased in cells acclimated to mixotrophy (Fig. 7e). The effect was substantial when calculated using an organelle distance of ≤ 30 nm (from 0.16 ± 0.3 to 1.8 ± 0.9) and still significant (twofold) at ≤ 90 nm (from 3.3 ± 1.7 to 6.9 ± 1.4). Although the proximity surface between the organelles is small, its increase could be relevant in the frame of the observed physiological changes. Plastid-mitochondria proximity may facilitate energy exchanges between the organelles in *Nannochloropsis*, to readjust the balance between the two cell organelles according to the environmental conditions. Alternatively, proximity could mediate lipid exchange between plastids and mitochondria (Mueller-Schuessele and Michaud, 2018), contributing to the structural changes observed between the two trophic conditions.

Typos:

Pg. 3: "cellular (subcellular) architecture" -> cellular and organellar architecture

done

Pg. 3: "2 dimensional (2D)" -> two dimensional (2D)

done

Pg. 3: "3-dimensional (3D)" -> three dimensional (3D)

done

Pg. 4: "phytoplankton representative" -> phytoplankton representatives

done

Pg. 14: "organelle interactions is an advantage" -> organelle interactions are an advantage

done

Reviewer #2 (Remarks to the Author):

I am fully satisfied with all changes made by the authors and recommend this manuscript for publication in Nature Communications.

Many thanks to referee 2 for the very positive comment.

Minor comment:

The authors may want to modify the sentence "Prior to FIB-SEM imaging, live cells were tested for photosynthetic capacity to verify their physiological status" (page 4) as the current wording is not precise enough regarding if multiple individual cells or culture aliquots (a mix of cells in different physiological states) were analyzed and what "physiological status" the cells had to have in order to be further processed for imaging.

We have modified the sentence to clarify that measurements were done using culture aliquots.

Page 4: Prior to FIB-SEM imaging, culture aliquots were tested for photosynthetic capacity (Supplementary Table 1) to verify their physiological status.